# Quantifying Statistical Significance of Deep Nearest Neighbor Anomaly Detection via Selective Inference

**Mizuki Niihori[1]** **Shuichi Nishino[1,2]** **Teruyuki Katsuoka[1]** **Tomohiro Shiraishi[1,2]**
**Kouichi Taji[1]** **Ichiro Takeuchi[1,2]**
**[1]Nagoya University** **[2]RIKEN**

niihori.mizuki.nagoyaml@gmail.com
takeuchi.ichiro.n6@f.mail.nagoya-u.ac.jp

## Abstract

In real-world applications, anomaly detection (AD) often operates without access to anomalous data, necessitating semi-supervised methods that rely solely on normal data. Among these methods, deep $k$-nearest neighbor (deep $k$NN) AD stands out for its interpretability and flexibility, leveraging distance-based scoring in deep latent spaces. Despite its strong performance, deep $k$NN lacks a mechanism to quantify uncertainty—an essential feature for critical applications such as industrial inspection. To address this limitation, we propose a statistical framework that quantifies the significance of detected anomalies in the form of $p$-values, thereby enabling control over false positive rates at a user-specified significance level (e.g.,0.05). A central challenge lies in managing selection bias, which we tackle using Selective Inference-a principled method for conducting inference conditioned on data-driven selections. We evaluate our method on diverse datasets and demonstrate that it provides reliable AD well-suited for industrial use cases.

## 1 Introduction

In many practical anomaly detection (AD) problems, anomalous data is often unavailable in advance; therefore, AD algorithms must be developed using only normal data-a setting known as semi-supervised AD [1–3]. Among various semi-supervised AD methods, we focus on the $k$-nearest neighbor ($k$NN) approach [4]. The $k$NN approach is simple yet effective, offering flexibility, minimal assumptions about the data, and adaptability to different distance metrics. Especially, by applying $k$NN in a latent feature space identified by deep learning models, anomalies can be detected more flexibly using task-specific distance metrics—an approach we refer to as *deep kNN*-based AD. Deep $k$NN-based AD combines the power of deep learning models with interpretable $k$NN scoring in latent space, making it a promising method for practical applications [5].

However, its dependence on complex deep learning-based detection procedures means that no established method currently exists for rigorously quantifying uncertainty or ensuring reliability. This limitation is particularly critical in high-stakes applications such as industrial inspection, where reliable uncertainty estimation is essential. Despite the interpretability of deep $k$NN-based scoring, it lacks a principled way to assess how confidently a test case deviates from normal cases. Therefore, developing a statistical framework that complements the high detection accuracy of deep $k$NN with uncertainty quantification is a crucial step toward practical and reliable deployment in real-world practical applications.

To address this issue, we propose a method that can quantify the statistical significance of the results obtained by deep $k$NN-based ADs. Specifically, we formulate the $k$NN-based AD as a statistical hypothesis testing problem and develop a method to quantify the statistical significance of detected anomalies in the form of $p$-values. The $p$-value of a detected anomaly represents the probability that the anomaly is a false positive. By making decisions based on anomalies with $p$-values below

39th Conference on Neural Information Processing Systems (NeurIPS 2025).

a certain threshold (e.g., 0.05), we can ensure that the probability of erroneous decisions remains below the significance level of 5%. In this study, we refer to the proposed testing method as the *deep-kNN test*. From a practical perspective, the ability to explicitly control the false positive rate is highly beneficial in real-world anomaly detection. In safety-critical applications such as medical diagnosis or industrial inspection, false positives often lead to unnecessary interventions, costs, or workflow disruptions. By providing a statistically significant upper bound on the false positive rate, our framework enables practitioners to make anomaly detection decisions with a quantifiable level of reliability.

Accurately quantifying the statistical significance of anomalies identified by deep $k$NN is a non-trivial challenging task from the following two perspectives. The first challenge arises from the fact that both AD and testing are performed on the same dataset, leading to selection bias called *double-dipping* [6]. The second challenge is that when anomalies are detected based on the latent feature space of a deep learning model, it becomes necessary to account for the complex process of the deep learning model computation. The core idea of our deep $k$NN test is to address these two challenges by introducing a statistical testing framework known as *Selective Inference (SI)* [7, 8]. SI has recently gained attention as a method for statistically testing data-driven hypotheses, by conditioning statistical inference on the event that the hypothesis has been selected. Figure 1 demonstrates how the proposed deep $k$NN test can be applied to industrial visual inspection task.

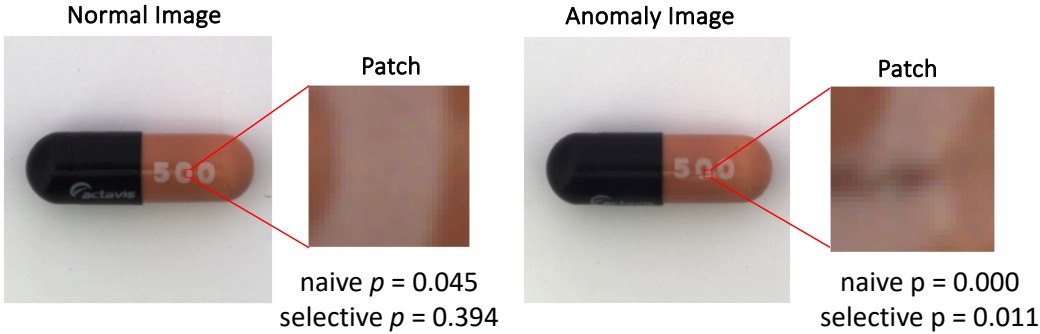

Figure 1: Examples of anomaly patches extracted from *Capsule* images using $k$NN-based AD are shown (see § 5 for detailed settings). For both the normal image (left) and the anomaly image (right), two types of $p$-values derived from different statistical tests are presented. The "naive $p$" represents the $p$-value obtained using a conventional method, while the "selective $p$" denotes the $p$-value computed using the method proposed in this study. At a significance level of $\alpha = 0.05$, the conventional naive $p$-value for the normal image (left) falls below the threshold, resulting in a false positive detection. In contrast, the proposed selective $p$-value correctly identifies it as a true negative. For the anomaly image (right), both $p$-values fall below the threshold, correctly identifying the patch as anomalous (true positive). In this study, we show that that conventional naive $p$-values are invalid as measures of statistical significance, whereas the proposed selective $p$-values serve as valid uncertainty measures for assessing the significance of anomalies detected by $k$NN-based ADs.

**Related Works**  Semi-supervised AD algorithms can generally be divided into three main categories [4, 9, 10]. The first group comprises AD methods based on parametric probabilistic models. These methods assume that normal data follow a specific statistical distribution, such as a multivariate Gaussian. A common traditional technique involves computing the Mahalanobis distance and assigning $p$-values using the $\chi^2$ distribution, allowing for statistical significance testing. These methods are interpretable and computationally efficient but may perform poorly with complex or non-Gaussian data distributions [11, 12]. The second group includes AD methods based on classical machine learning algorithms. Representative methods include One-Class SVM [13], Isolation Forest [14], and k-Nearest Neighbors ($k$NN) [15]. While these methods are more flexible than parametric probabilistic model-based methods, they often lack a principled mechanism to quantify statistical significance of the detected anomalies, making it difficult to assess confidence in AD. The third group comprises deep learning-based AD methods, which use neural networks to capture complex patterns in normal data. Representative approaches include autoencoders or variational autoencoders (VAEs) [16], GAN-based method [17], and diffusion-based method [18]. Deep SVDD [1] and deep

$k$NN-based AD [19] are also widely used as AD methods based on deep learning models. While these deep learning-based approaches often achieve high detection performance, most still lack a well-established framework for quantifying statistical significance, such as assigning $p$-values.

Selective Inference (SI) -also known as post-selection inference- provides valid statistical inference for data-driven hypotheses. By leveraging the conditional distribution of a test statistic given that a particular hypothesis was selected based on the data, SI corrects the selection bias introduced by *cherry-picking* findings. This ensures that the reported $p$-values -known as selective $p$-values- maintain valid false positive rates, even when the same data is used for both hypothesis selection and testing, which would otherwise inflate the error rates due to *double dipping* [7]. Early work in SI primarily focused on feature selection in linear regression. A seminal contribution by Lee et al [8] introduced an exact SI procedure for the Lasso, deriving valid selective $p$-values for selected regression coefficients by conditioning on the selection event induced by the Lasso solution. Since then, extensive research has extended SI to various other feature selection settings, including marginal screening [20], stepwise feature selection [21], generalized linear models [22], and many others [23–33]. Recent developments have also focused on improving the power of SI methods through new theoretical insights and algorithmic innovations [34–38]. In parallel, SI has been adapted to problems beyond feature selection, finding applications in a range of domains, such as clustering [39–42] and many others [43–47]. In the context of deep learning, SI has been applied to provide statistical inference for segmentation tasks [48], saliency maps such as CAM [49], and attention weights in Vision Transformers [50]. In recent years, this line of research has begun to explore the reliability of various deep learning model components from a statistical inference perspective [51–53]. In contrast, applications of SI in AD settings remain limited. One related work is the application of SI to change-point detection in time series data [54–58]. However, that line of research focuses on testing the significance of global changes in the entire sequence, which differs from our setting where the goal is to assess the abnormality of individual data points. Another related work is the application of SI to robust regression [59], where inference is performed on regression coefficients obtained after excluding outliers. However, the statistical significance of the detected outliers themselves is not addressed. In conclusion, no studies have yet explored the SI framework to quantify the statistical significance of $k$NN-based AD, and its deep $k$NN variants remain entirely unexplored, leaving a significant gap in the literature.

**Our contributions**  In this paper, as a proof of concept, we address the problem of quantifying the statistical significance of results produced by a simple deep $k$NN-based anomaly detection (AD) approach, as studied in [19], where anomalies are identified by thresholding the $k$NN distance in the latent feature space of a CNN classifier. Unlike prior studies on deep $k$NN-based AD, our contribution is *not* the development of a new AD algorithm aimed at improving detection accuracy, but rather the introduction of a method to quantify the statistical uncertainty of detected anomalies. Our first contribution is to formulate $k$NN-based semi-supervised AD as a statistical test within the SI framework, enabling rigorous reliability assessment of detected anomalies. The second contribution is to develop a computational method to incorporate the distance measure derived from a deep learning model into the SI framework. Finally, we validate the effectiveness of the proposed deep-$k$NN test through experiments on various datasets and industrial inspection scenarios, demonstrating its practical utility and robustness. Furthermore, to facilitate reproducibility and further research, we release an open-source implementation of our proposed method, including code and experimental scripts, available at `https://github.com/Takeuchi-Lab-SI-Group/Quantifying_Statistical_Significance_of_Deep_Nearest_Neighbor_Anomaly_Detection_via_SI`.

## 2 Problem setup: deep $k$NN-based anomaly detection

In this section, we describe the problem setting of a simple deep $k$NN-based AD, nearly identical to that in [19], as a proof of concept for our statistical testing framework. In semi-supervised AD problems, the available training dataset consists only of the set of normal instances. Let $\boldsymbol{x}_1, \ldots, \boldsymbol{x}_n \in \mathbb{R}^d$ represent the set of $d$-dimensional input feature vectors for $n$ normal training instances, where $n$ is the number of instances. If an instance is an image, for example, $d$ is the number of pixels in the image and $\boldsymbol{x}_i, i \in [n]$ is the $d$-dimensional vector of pixel values. We assume that a pretered CNN is available, and let us denote its feature representation as $\boldsymbol{\phi}(\boldsymbol{x}_i), i \in [n]$ using a feature extractor $\boldsymbol{\phi} : \mathbb{R}^d \to \mathbb{R}^D$, where $D$ is the dimension size of the latent feature vectors. We measure the distance

between two input instances $\boldsymbol{x}, \boldsymbol{x}' \in \mathbb{R}^d$ in the latent feature space of a pre-trained CNN classifier as

$$\mathrm{dist}_{\boldsymbol{\phi}}(\boldsymbol{x}, \boldsymbol{x}') := \|\boldsymbol{\phi}(\boldsymbol{x}) - \boldsymbol{\phi}(\boldsymbol{x}')\|_2, \tag{1}$$

where we adopt the $L_2$ distance within the latent feature space in this study for concreteness, although the choice of distance metric is flexible and discussed further in the §6.

Given a test instance $\boldsymbol{x}^{\text{test}} \in \mathbb{R}^d$, the $k$NN-based AD is formulated as follows. Consider an order of indices by ascending distance with respect to a distance function $\mathrm{dist}_{\phi}(\cdot, \cdot)$ such that

$$\mathrm{dist}_{\phi}(\boldsymbol{x}^{\text{test}}, \boldsymbol{x}_{o(1)}) \leq \mathrm{dist}_{\phi}(\boldsymbol{x}^{\text{test}}, \boldsymbol{x}_{o(2)}) \leq \cdots \leq \mathrm{dist}_{\phi}(\boldsymbol{x}^{\text{test}}, \boldsymbol{x}_{o(n)}). \tag{2}$$

Then, $\boldsymbol{x}_{o(k)}$ is called the $k^{\text{th}}$ nearest neighbor instance of $\boldsymbol{x}^{\text{test}}$. Since the choice of $k$ affects the distance magnitude, we adopt the following well-known anomaly score [4, 60]:

$$a(\boldsymbol{x}^{\text{test}}) = \log \mathrm{dist}\left(\boldsymbol{x}^{\text{test}}, \boldsymbol{x}_{o(k)}\right) - \frac{\log k}{D}, \tag{3}$$

where the first term represents the log-scale distance, whereas the second term adjusts for the influence of $k$'s selection[1]. In $k$NN-based AD, if Eq. (3) exceeds a certain threshold $\theta$, the test instance $\boldsymbol{x}^{\text{test}}$ is selected as an anomaly. The threshold $\theta$ is typically determined based on the empirical distribution of anomaly scores among normal instances. The choice of $k$ greatly affects the results in $k$NN-based AD. Users can set $k$ based on domain knowledge or experience. However, when domain knowledge is limited or data is complex, a systematic approach is desirable. In semi-supervised AD, unlike supervised learning such as $k$NN classification or regression, it is not possible to determine $k$ through data splitting such as cross-validation. One commonly used heuristic to select $k$ is to calculate the anomaly scores for various $k$ values per test instance $\boldsymbol{x}^{\text{test}}$, choosing the $k$ that maximizes this score. Our proposed $k$NN-test is valid regardless of the method used to determine $k$.

## 3 Statistical testing framework for $k$NN-base AD

In this section, we formulate the $k$NN-based AD problem as a statistical hypothesis testing problem in order to quantify the statistical significance of the detected anomalies. First, we consider each input feature vector $\boldsymbol{x}_i, i \in [n]$ as a realization of a random vector $\boldsymbol{X}_i, i \in [n]$. The statistical model for the random vector $\boldsymbol{X}_i$ is written as

$$\boldsymbol{X}_i = \boldsymbol{s}_i + \boldsymbol{\varepsilon}_i, \quad i \in [n], \tag{4}$$

where $\boldsymbol{s}_i \in \mathbb{R}^d$ represents the signal component, and $\boldsymbol{\varepsilon}_i \in \mathbb{R}^d$ represents the noise component. In this study, we adopt a *semi-parametric* assumption for the statistical model described in Eq.(4). Specifically, we place no restrictions on the distribution of the signal components $\boldsymbol{s}_i$ for $i \in [n]$, treating them in a completely non-parametric fashion. In contrast, we assume that the noise component follows a Gaussian distribution $\mathcal{N}(\boldsymbol{0}, \sigma^2 I)$, where $\sigma^2$ is either known or can be estimated from an independent dataset. This setup differs from traditional AD approaches based on parametric probabilistic models, which assume that the signal components follow a specific parametric distribution. In our semi-parametric framework, the distribution of the signal components is entirely unknown and unrestricted, allowing the model to remain valid even when the signals exhibit complex or multimodal characteristics. For example, if the input feature vectors $\boldsymbol{x}_i$ represent images—each element corresponding to a pixel value—then our model in Eq.(4) allows a set of arbitrary original images each of which is contaminated by Gaussian noise. Our goal is to determine, within statistical hypothesis testing framework, whether an observed anomaly originates from the underlying signal or is merely a consequence of the noise.

Let the feature vector of a test instance be denoted as $\boldsymbol{x}^{\text{test}}$ and its corresponding random version as $\boldsymbol{X}^{\text{test}}$. We assume $\boldsymbol{X}^{\text{test}} = \boldsymbol{s}^{\text{test}} + \boldsymbol{\varepsilon}^{\text{test}}$ in the same way as Eq.(4). In $k$NN-based AD, the $k$-nearest instance $\boldsymbol{x}_{o(k)}$ is selected from the $n$ available training instances. Our interest in the test is to determine whether the signal at the representative instance of $k$ nearest neighbor points $\mathcal{K} = \{\boldsymbol{X}_{o(1)}, \boldsymbol{X}_{o(2)}, \cdots, \boldsymbol{X}_{o(k)}\}$ is statistically different from the signal of the test instance.

$$\boldsymbol{x}_{\mathrm{med(k)}} := \underset{\{\boldsymbol{x}_j | j \in \mathcal{K}\}}{\arg\min} \sum_{i \in \mathcal{K}} \mathrm{dist}_{\phi}(\boldsymbol{x}_i, \boldsymbol{x}_j), \tag{5}$$

---

[1]The choice of Eq. (3) is based on certain assumptions and heuristics in the literature, but its details are beyond the scope of this paper. For further information, refer to [4].

and denote its corresponding random version as $\boldsymbol{X}_{\text{med}(k)}$. This problem can be formulated as a hypothesis testing problem using the following null hypothesis $H_0$ and alternative hypothesis $H_1$:

$$H_0 : \boldsymbol{s}^{\text{test}} = \boldsymbol{s}_{\text{med}(k)} \quad \text{vs.} \quad H_1 : \boldsymbol{s}^{\text{test}} \neq \boldsymbol{s}_{\text{med}(k)} \tag{6}$$

The null hypothesis $H_0$ states that the true signal of the $k$-medoid normal training instance equals the true signal of the test instance, while the alternative hypothesis $H_1$ asserts they are different. By performing a statistical test for these hypotheses, the false detection rate of an anomaly can be quantified using $p$-values.

As a reasonable test statistic, we consider

$$T\left(\boldsymbol{X}^{\text{test}}, \{\boldsymbol{X}_i\}_{i \in [n]}\right) := \frac{1}{\sqrt{2}} \left\|\boldsymbol{X}^{\text{test}} - \boldsymbol{X}_{\text{med}(k)}\right\|_2. \tag{7}$$

The $p$-value is defined as the probability of observing a test statistic greater than or equal to the one actually observed under the null hypothesis $H_0$, i.e.,

$$p = \mathbb{P}_{H_0}\left(T\left(\boldsymbol{X}^{\text{test}}, \{\boldsymbol{X}_i\}_{i \in [n]}\right) \geq T\left(\boldsymbol{x}^{\text{test}}, \{\boldsymbol{x}_i\}_{i \in [n]}\right)\right). \tag{8}$$

Unfortunately, the probability in Eq. (8) is computationally intractable, as it depends on the complex computation process of deep $k$NN. To address this issue, we introduce the SI framework and define a significance quantification measure called the *selective p-values*. In the following section, we present the concept of selective $p$-values and demonstrate that this measure can appropriately quantify the statistical significance of anomalies detected by $k$NN-based AD.

## 4    Selective Inference (SI) for $k$NN-base AD

In this section, we propose selective $p$-values, based on the framework of SI, as a measure of the statistical significance of anomalies detected by $k$NN-based AD. These selective $p$-values can be interpreted in the same way as conventional $p$-values. For example, if the significance level is set to $\alpha = 0.05$ and we consider anomalies with selective $p$-values less than 0.05, it is theoretically guaranteed that the proportion of falsely detected anomalies will remain below 0.05.

### 4.1    Alternative formulations of the test statistic in Eq.(7)

First, let us denote the $(1 + n)d$-dimensional vector obtained by concatenating the test instance $\boldsymbol{x}^{\text{test}}$ and $n$ training instance $\boldsymbol{x}_1, \ldots, \boldsymbol{x}_n$, all of which are $d$-dimensional vectors, as

$$\boldsymbol{y} = \text{vec}\left(\boldsymbol{x}^{\text{test}}, \boldsymbol{x}_1, \ldots, \boldsymbol{x}_n\right) \in \mathbb{R}^{(1+n)d}, \tag{9}$$

where $\text{vec}$ is the operation that concatenates multiple vectors into a single column vector. Similarly, the $(1 + n)d$-dimensional vector obtained by concatenating $1 + n$ random vectors is denoted as

$$\boldsymbol{Y} = \text{vec}\left(\boldsymbol{X}^{\text{test}}, \boldsymbol{X}_1, \ldots, \boldsymbol{X}_n\right) \in \mathbb{R}^{(1+n)d}. \tag{10}$$

With these notations, we can rewrite the test statistic in Eq.(7) as

$$T(\boldsymbol{Y}) = \|P\boldsymbol{Y}\|_2, \tag{11}$$

where $P \in \mathbb{R}^{(1+n)d \times (1+n)d}$ is a symmetric block matrix with each block of size $(d \times d)$, defined as follows:

$$P = \frac{1}{2} \begin{pmatrix} I_d & O & \cdots & -I_d & \cdots & O \\ O & O & \cdots & O & \cdots & O \\ \vdots & \vdots & \ddots & \vdots & & \vdots \\ -I_d & O & \cdots & I_d & \cdots & O \\ \vdots & \vdots & & \vdots & \ddots & \vdots \\ O & O & \cdots & O & \cdots & O \end{pmatrix} \tag{12}$$

Note that the only non-zero blocks are the identity matrix $I_d$ in $((0, 0))$, $((\text{med}(k) + 1, \text{med}(k) + 1))$ and $-I_d$ in $((0, \text{med}(k) + 1))$, $((\text{med}(k) + 1, 0))$, and all other blocks are zero matrices.

## 4.2 Naive $p$-values

Here, we discuss a measure referred to as the *naive p-value*. Although this measure is *invalid* as an indicator of statistical significance, it serves as a contrasting concept that helps introduce the notion of selective $p$-values. The naive $p$-value is defined as

$$p_{\mathrm{naive}} := \mathbb{P}_{\mathrm{H}_0}\left(\|PY\|_2 \geq \|Py\|_2\right),\tag{13}$$

where we emphasize the distinction between the random vector $Y$ and the observed vector $y$. Under the statistical model in Eq.(4), the random vector $Y$ follows a multivariate normal distribution. Therefore, the statistic $\tilde{T}(Y)$ follows a $\chi$ distribution with $d$ degrees of freedom. Consequently, Eq.(13) can be easily computed as the tail probability of $\chi(d)$ distribution. Unfortunately, this easily computable naive $p$-value is invalid in the sense that it does not account for the fact that the $k$-nearest neighbor instance $x_{o(k)}$ is selected based on the same observed data. If naive $p$-values are used for decision-making in the same way as ordinary $p$-values, the false detection rate cannot be properly controlled as intended.

## 4.3 Selective $p$-values

The basic idea of SI, pioneered by the seminal work by Lee et al. [8], is to address the problem of the naive $p$-values by employing the framework of conditional testing, based on the key insight that the sampling distribution of a test statistic can be tractable if the statistic is conditioned on.

In order to define selective $p$-value, let us represent an event that the $k$-nearest neighbor index $o(k)$ is selected based on the random vector $Y$ as "$\mathcal{E}_Y = o(k)$". Then, selective $p$-value is defined as

$$p_{\mathrm{selective}} = \mathbb{P}_{\mathrm{H}_0}\left(\|PY\|_2 \geq \|Py\|_2 \mid \mathcal{E}_Y = \mathcal{E}_y, \mathcal{Q}_Y = \mathcal{Q}_y\right),\tag{14}$$

where the first condition "$\mathcal{E}_Y = \mathcal{E}_y$" indicates the event that the $k$-nearest neighbor index $o(k)$ obtained from the random data vector $Y$ is the same as that obtained from the observed data vector $y$. The second condition "$\mathcal{Q}_Y = \mathcal{Q}_y$" indicates that the sufficient statistic of the nuisance parameter defined as

$$\mathcal{Q}_Y = \left(\frac{PY}{\|PY\|}, \left(I_{(n+1)d} - P\right)Y\right),\tag{15}$$

where $P = \eta_y \eta_y^\top \in \mathbb{R}^{(1+n)d \times (1+n)d}$, is the same for both the random data vector $Y$ and the observed data vector $y$. Here, the key idea is, by the first conditioning on the $k$-nearest neighbor index $o(k)$, $\eta_Y$ is fixed as $\eta_y$, making the computation of the probability in Eq.(14) tractable with the use of $\chi$ distribution as with the case of the naive $p$-value. Due to space limitations, we omit the mathematical details and statitical rationale of SI, including the role of the event related to the nuisance parameter [2]. For further details, we refer the reader to the literature such as [7, 8, 34].

The computation of selective $p$-value in Eq.(14) is reduced to a tail probability computation of *truncated $\chi$ distribution* as formally stated in the following theorem.

**Theorem 4.1.** *The following conditional test statistic*

$$\|PY\|_2 \mid \{\mathcal{E}_Y = \mathcal{E}_y, \mathcal{Q}_Y = \mathcal{Q}_y\}\tag{16}$$

*follows a truncated $\chi$ distribution with $(1+n)d$ degrees of freedom, where the truncation is determined by the constraint "$\mathcal{E}_Y = \mathcal{E}_y$" and the domain of the distribution is on the one-dimensional subspace defined by $\{Y \mid \mathcal{Q}_Y = \mathcal{Q}_y\}$.*

The proof of Theorem 4.1 is deferred to Appendix A.2. Furthermore, the selective $p$-value in Eq.(14) is valid $p$-value in the sense that, for any significance level $\alpha \in (0,1)$, the false detection rate of the anomalies with $p_{\mathrm{selective}} < \alpha$ is exactly $\alpha$ as formally stated in the following theorem.

---

[2]In this context, the nuisance parameter refers to a parameter that is included in the null distribution but is not of direct inferential interest. To characterize the null distribution, it is necessary to eliminate the influence of the nuisance parameter. In our approach, this is achieved by conditioning on its sufficient statistic. This is a standard technique in the field of selective inference [7, 8, 34].

**Theorem 4.2.** *The selective $p$-values defined in Eq.(14) satisfies*

$$\mathbb{P}_{\mathrm{H}_0}\left(p_{\mathrm{selective}} \leq \alpha\right) = \alpha, \ \forall \alpha \in (0, 1). \tag{17}$$

The proof of Theorem 4.2 is deferred to Appendix A.2. The statement of Theorem 4.2 indicates that the selective $p$-value in Eq.(14) can be used as a measure to quantify the statistical significance of the detected anomalies.

### 4.4 Selection event characterization

The main technical challenge in computing selective $p$-values lies in characterizing the selection event "$\mathcal{E}_{\boldsymbol{Y}} = \mathcal{E}_{\boldsymbol{y}}$". Conditioning on this selection event means identifying the set of vectors $\boldsymbol{Y} \in \mathbb{R}^{(1+n)d}$ such that the same normal instance $\boldsymbol{x}_{o(k)}$, which was selected as the $k$-nearest neighbor based on the observed data vector $\boldsymbol{y}$, is also selected as the $k$-nearest neighbor. To realize this, one must appropriately characterize and account for several factors, including the computation of latent feature vectors in the trained CNN model and the comparison of distances between the test instance and normal instances in the latent feature space. Due to space limitations, the detailed characterization of the selection event is provided in Appendix B, where we adopt the parametric programmingbased approach proposed in [38] as the core computational framework.

## 5 Numerical Experiments

In this section, we demonstrate that the proposed method exhibits high power (true positive rate) while controlling the Type I error rate (false positive rate) below the significance level compared to other methods. First, experiments are conducted on synthetic datasets, followed by similar experiments on two types of real datasets. All experiments are conducted with a significance level $\alpha = 0.05$. We examined the case where $\phi$, a mapping to a latent feature space, is the identity function in simple settings for the synthetic and tabular data experiments, while for the image data experiments, we considered the case where $\phi$ is defined by a DNN model. We executed all experiments on AMD EPYC 9474F processor, 48-core 3.6GHz CPU and 768GB memory.

### 5.1 Baseline Methods

In the experiments on synthetic datasets and tabular datasets, we compared the proposed method (`proposed`) with four other baseline methods: `Hotelling_t2`, `w/o-pp`, `naive`, and `Bonferroni`. Subsequently, in the experiments on image datasets, we additionally compare two further ablation studies: `OpA1` and `OpA2`.

- `Hotelling_t2`: This method is not based on $k$NN but instead performs classical anomaly detection by computing a $p$-value using Hotellings $T^2$ test. Unlike `proposed`, which assumes semi-parametric conditions on the training data as shown in Eq.(4), this method makes a stricter parametric assumption that all training samples are i.i.d. from a single Gaussian distribution. It uses the squared Mahalanobis distance $T^2(\boldsymbol{X}^{\mathrm{test}})$ as the test statistic and computes the $p$-value as $p_{\mathrm{hotelling}} = \mathbb{P}_{\mathrm{H}_0}(T^2(\boldsymbol{X}^{\mathrm{test}}) \geq T^2(\boldsymbol{x}^{\mathrm{test}}))$, where the null distribution is assumed to follow the $\chi^2$ distribution.

- `Bonferroni`: We chose the `Bonferroni` correction as the most basic multiple comparison method[3]. This is a method to control the Type I error rate by using the Bonferroni correction. There are $\binom{n}{k}$ ways to choose the neighbors $\mathcal{N}$, then we compute the Bonferroni corrected $p$-value as $p_{\mathrm{bonferroni}} = \min(1, \binom{n}{k} \cdot p_{\mathrm{naive}})$.

- `naive`: This is a statistical test that uses the naive $p$-value defined in Eq.(13).

- `w/o-pp`: An ablation study that excludes the parametric programming technique described in Appendix B.

- `OpA1`: Another ablation study that excludes the selection events for $k$NN (i.e., $\mathcal{M}_{\boldsymbol{Y}}, \mathcal{N}_{\boldsymbol{Y}}, \mathcal{K}_{\boldsymbol{Y}}$, and $\mathcal{S}_{\boldsymbol{Y}}$ in Appendix B).

- `OpA2`: Another ablation study that excludes the selection events for DNN (i.e., $\mathcal{D}_{\boldsymbol{Y}}$ in Appendix B).

---

[3]Other multiple testing procedures, such as Holm's method, require computing nominal $p$-values for all possible hypotheses, which is an intractable task due to the combinatorial explosion.

## 5.2 Synthetic Datasets

To evaluate the Type I error rate, we varied the training dataset size $n \in \{100, 200, 500, 1000\}$ and set the data dimension $d = 5$. The number of neighbors $k$ was adaptively selected in a data-driven manner from $\{1, 2, 5, 10\}$. See Appendix C.2 for results when $d$ and $k$ are varied. For each configuration, we conducted 1,000 independent experiments. In each iteration, we generated a test instance $\boldsymbol{x}^{\text{test}}$ and training instances $\boldsymbol{x}_i$ for $i \in [n]$, sampled from the same distribution under one of the following two settings. In the **parametric** setting, all instances were drawn from a standard Gaussian distribution $\mathcal{N}(\boldsymbol{0}, \boldsymbol{I}_d)$. In the **semi-parametric** setting, each instance was drawn from $\mathcal{N}(\boldsymbol{s}_i, \boldsymbol{I}_d)$, where $\boldsymbol{s}_i$ is a randomly generated mean vector. For more details about data generation, see Appendix C.1. To evaluate the power, we generated data in the same way, except that a signal $\delta \in \{2, 4, 6, 8\}$ was added to a randomly selected coordinate of the test instance $\boldsymbol{x}^{\text{test}}$. We set $d = 5$, $n = 100$ and $k$ was adaptively selected in the same way. See Appendix C.3 for results when $d$, $k$, and $n$ are varied. The results of Type I error rate are shown in Figure 2. The `proposed`, `w/o-pp`, and `Bonferroni` successfully controlled the Type I error rate under the significance level, whereas the `naive` could not. Since `naive` failed to control the Type I error rate, we excluded it from the power experiment. `Hotelling_t2` also fails in the semi-parametric setting (where $s_i$ follows a non-Gaussian distribution), because it assumes that all samples are i.i.d. from a single Gaussian distribution. So, it is likewise excluded from the power experiment in the semi-parametric setting. The results of power are shown in Figure 3. Among the methods that controlled the Type I error rate, the `proposed` has the highest power.

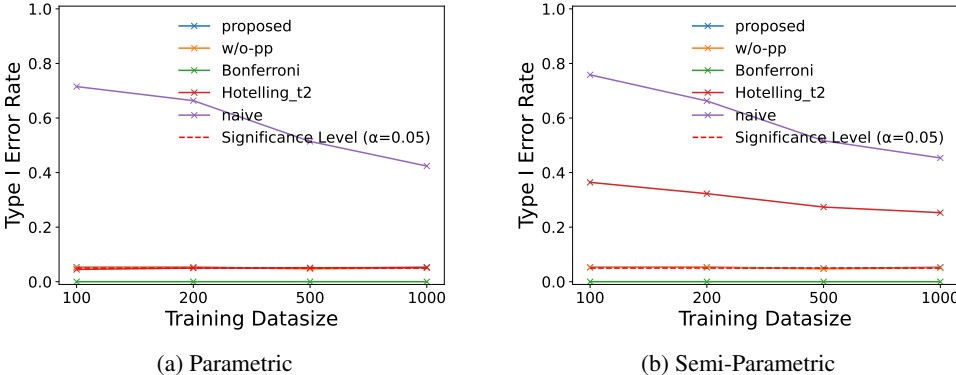

|              |                   |
|:------------:|:-----------------:|
| (a) Parametric | (b) Semi-Parametric |

Figure 2: Results of Type I error rate when changing the dataset size $n$. `proposed`, `w/o-pp`, and `Bonferroni` successfully control the Type I error rate across all settings. Their lines are almost overlapping. `naive` fails and the results of `Bonferroni` are almost zero, because it is too conservative. `Hotelling_t2` also fails in the semi-parametric setting.

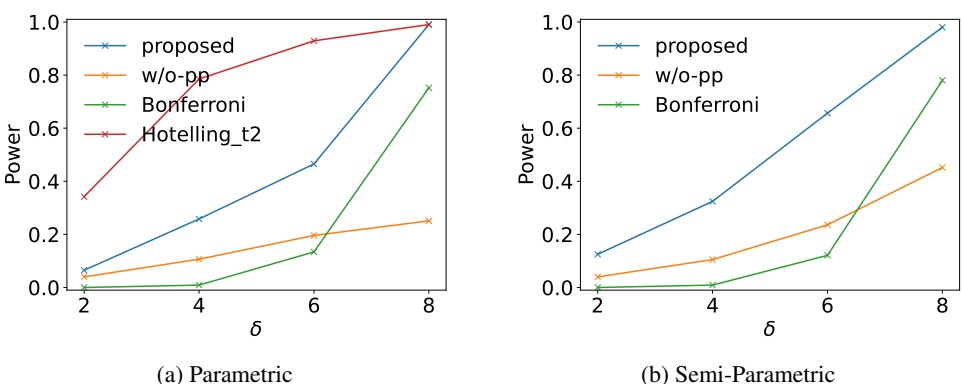

|              |                   |
|:------------:|:-----------------:|
| (a) Parametric | (b) Semi-Parametric |

Figure 3: Power when varying signal strength $\delta$. `proposed` and `Hotelling_t2` outperformed other methods. However, `Hotelling_t2` failed to control the Type I error rate in the semi-parametric setting so it is not shown in the power results.

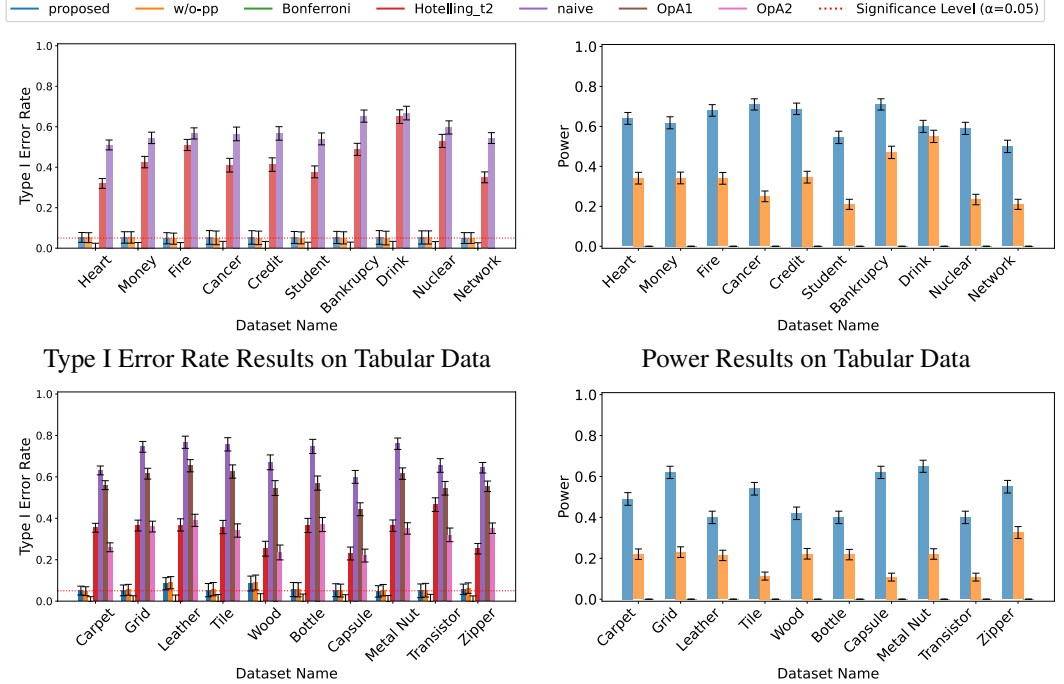

Type I Error Rate Results on Tabular Data          Power Results on Tabular Data

Type I Error Rate Results on Image Data          Power Results on Image Data

Figure 4: Results on real datasets. The proposed method (`proposed`) outperformed the other methods in terms of power, while controlling the Type I error rate below the significance level across all datasets. The Type I error rate, power, and error bars of the `Bonferroni` are almost zero, because it is too conservative.

## 5.3 Real Datasets I: Tabular Data

We conducted evaluations using 10 tabular real-world datasets. These datasets reflect various real-world problems from different domains. The datasets used in our experiments are listed in Appendix C.4. Only numerical features from each dataset were used in the experiments. The datasets vary in dimensionality, ranging from 4 to 10 dimensions. The number of neighbors $k$ was adaptively selected in a data-driven manner from $\{1, 2, 5, 10\}$. Before conducting the experiments, All datasets are standarized with each feature having mean 0 and variance 1. The results of the Type I error rate and power are shown in Figure 4. The `proposed` method outperformed the other methods in terms of power, while controlling the Type I error rate.

## 5.4 Real Datasets II: Image Data

In this experiment, we used the MVTec AD dataset [61, 62]. The dataset consists of 15 classes, and we chose 10 classes for the experiments which seem to follow a normal distribution. The datasets used in our experiments are listed in Appendix C.5. Before conducting the experiments, All datasets are standarized with each feature having mean 0 and variance 1. Following the conventional Deep $k$NN approach [19], we employed a ResNet model pre-trained on the ImageNet dataset as a feature extractor in this experiment. As a preprocessing step, the original image, which has a size of $900 \times 900$ or $1024 \times 1024$, was divided into $30 \times 30$ patches, and the patch was used as the test instance. For the training instances, we used 100 patches from the same position as the test instance. We set the number of neighbors $k = 3$. The results of the Type I error rate and power are shown in Figure 4. The `proposed` method outperformed the other methods in terms of power, while controlling the Type I error rate below the significance level. Some examples of the experimental results are shown in Figure 5 and Figure 15.

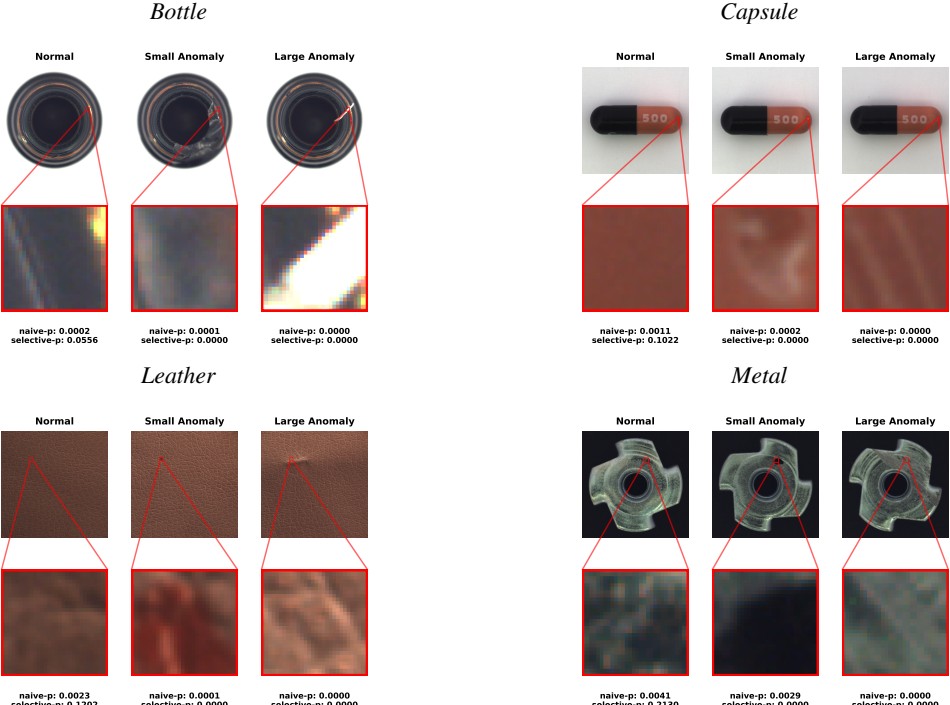

Figure 5: Experimental results of 4 datasets from MVTec AD dataset. For each dataset, one normal example (left) and two anomaly examples (center, right) are showed. For each example, the top row displays the original image used for testing along with the patch location (marked in red), while the bottom row presents the extracted patch image. For all normal examples, the naive $p$-value is below the significance level $\alpha = 0.05$ (false positive), whereas the proposed selective $p$-value correctly results in a true negative. For all anomaly examples, the selective $p$-value successfully detects anomalies.

## 6   Scope, limitations and conclusions

In this study, we proposed a method for quantifying uncertainty in $k$NN-based AD by assessing statistical significance. Uncertainty quantification in the outputs of deep learning models remains a major challenge in machine learning community, and our work contributes toward addressing this gap. This is particularly important for AD because it is often used in high-stakes applications such as medical diagnosis and industrial inspection, where evaluating statistical significance is crucial for practical reliability. Therefore, our proposed method, which enables the quantification of statistical significance of the detected anomalies, has substantial practical importance. Our SI-based fundamental idea is not limited to the specific $k$NN-based AD algorithm presented as a proof of concept in this study; it is also applicable to various other variants. For SI of deep $k$NN-based AD, it is essential to characterize both the computations by the deep learning model and the selection event of the $k$NN instances. The former is applicable to many CNN-type networks, while the latter applies to distance metrics such as $L_1$, $L_2$, and $L_\infty$. It is, in principle, extendable to other deep AD methods, as long as the selection event (i.e., $\mathcal{E}_Y = \mathcal{E}_y$ in Eq.(14)) can be characterized in a tractable form.

One limitation of the proposed $k$NN test lies in the form of the semi-parametric model discussed in Section 3. In this model, the signal components are entirely non-parametric, offering significantly greater flexibility than conventional statistical AD. However, the need to assume a distribution for the noise component remains a limitation. Additional experimental results on robustness when the noise distribution deviates from the normal distribution are presented in Appendix C, in which we observe that when the deviation is small, the false detection rate can be maintained at approximately the desired level. Another limitation arises when the selection event becomes more complex, rendering the current framework inapplicable in its existing form. This issue, for instance, occurs when a

Transformer is used as the deep learning model for identifying the latent feature space. This remains an important challenge for our future work.

## Acknowledgments and Disclosure of Funding

This work was partially supported by JST CREST (JPMJCR21D3, JPMJCR22N2), JST Moonshot R&D (JPMJMS2033-05), RIKEN Center for Advanced Intelligence Project, and JSPS KAKENHI Grant Number JP24K15080.

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

# A  Proof of Theorems

## A.1  Proof of Theorem 4.1

According to the condition on $\mathcal{E}_{\boldsymbol{Y}} = \mathcal{E}_{\boldsymbol{y}}$ and $\mathcal{Q}_{\boldsymbol{Y}} = \mathcal{Q}_{\boldsymbol{y}}$, i.e., $\mathcal{U}(\boldsymbol{Y}) = \mathcal{U}(\boldsymbol{y})$ and $\mathcal{V}(\boldsymbol{Y}) = \mathcal{V}(\boldsymbol{y})$, where

$$\mathcal{V}(\boldsymbol{Y}) = \frac{P\boldsymbol{Y}}{\|P\boldsymbol{Y}\|_2} \in \mathbb{R}^{(n+1)d}, \quad \mathcal{U}(\boldsymbol{Y}) = (I_{(n+1)d} - P)\boldsymbol{Y} \in \mathbb{R}^{(n+1)d}.$$

we have

$$\mathcal{U}(\boldsymbol{Y}) = \mathcal{U}(\boldsymbol{y})$$
$$\Leftrightarrow (I_{(n+1)d} - P)\boldsymbol{Y} = \mathcal{U}(\boldsymbol{y})$$
$$\Leftrightarrow \boldsymbol{Y} = \mathcal{U}(\boldsymbol{y}) + \mathcal{V}(\boldsymbol{Y})z$$
$$\Leftrightarrow \boldsymbol{Y} = \mathcal{U}(\boldsymbol{y}) + \mathcal{V}(\boldsymbol{y})z \quad (\because \mathcal{V}(\boldsymbol{Y}) = \mathcal{V}(\boldsymbol{y}))$$
$$\Leftrightarrow \boldsymbol{Y} = \boldsymbol{a} + \boldsymbol{b}z,$$

where $\boldsymbol{a} = \mathcal{U}(\boldsymbol{y}), \boldsymbol{b} = \mathcal{V}(\boldsymbol{y})$, and $z = T(\boldsymbol{Y}) = \|P\boldsymbol{Y}\|_2$.

Then, we have

$$\{\boldsymbol{Y} \in \mathbb{R}^{(1+n)d} \mid \mathcal{E}_{\boldsymbol{Y}} = \mathcal{E}_{\boldsymbol{y}}, \mathcal{Q}(\boldsymbol{Y}) = \mathcal{Q}(\boldsymbol{y})\}$$
$$= \{\boldsymbol{Y} \in \mathbb{R}^{(1+n)d} \mid \mathcal{E}_{\boldsymbol{Y}} = \mathcal{E}_{\boldsymbol{y}}, \boldsymbol{Y} = \boldsymbol{a} + \boldsymbol{b}z, z \in \mathbb{R}\}$$
$$= \{\boldsymbol{Y} = \boldsymbol{a} + \boldsymbol{b}z \in \mathbb{R}^{(1+n)d} \mid \mathcal{E}_{\boldsymbol{a}+\boldsymbol{b}z} = \mathcal{E}_{\boldsymbol{y}}, z \in \mathbb{R}\}$$
$$= \{\boldsymbol{Y} = \boldsymbol{a} + \boldsymbol{b}z \in \mathbb{R}^{(1+n)d} \mid z \in \mathcal{Z}\},$$

where $\mathcal{Z}$ is the truncation region defined as

$$\mathcal{Z} = \{z \in \mathbb{R} \mid \mathcal{E}_{\boldsymbol{a}+\boldsymbol{b}z} = \mathcal{E}_{\boldsymbol{y}}\}.$$

Therefore, by noting that $\|P\boldsymbol{s}\|_2$ is zero, we obtain

$$T(\boldsymbol{Y}) \mid \{\mathcal{E}_{\boldsymbol{Y}} = \mathcal{E}_{\boldsymbol{y}}, \mathcal{Q}(\boldsymbol{Y}) = \mathcal{Q}(\boldsymbol{y})\} \sim \mathrm{TC}(\mathrm{tr}(P), \mathcal{Z}),$$

where $\mathrm{TC}(\mathrm{tr(P)}, \mathcal{Z})$ is a truncated $\chi$-distribution with the degrees of freedom $d$, whose domain is the truncation region $\mathcal{Z}$.

## A.2  Proof of Theorem 4.2

The sampling distribution of the test statistic conditional on $\mathcal{E}_{\boldsymbol{Y}} = \mathcal{E}_{\boldsymbol{y}}$ and $\mathcal{Q}(\boldsymbol{Y}) = \mathcal{Q}(\boldsymbol{y})$ denoted by

$$T(\boldsymbol{Y}) \mid \{\mathcal{E}_{\boldsymbol{Y}} = \mathcal{E}_{\boldsymbol{y}}, \mathcal{Q}(\boldsymbol{Y}) = \mathcal{Q}(\boldsymbol{y})\}$$

is a truncated $\chi$-distribution with the degrees of freedom $(1+n)d$ and the truncation region $\mathcal{Z}$ defined in Theorem 4.1. Thus, by applying the probability integral transform, under the null hypothesis,

$$p_{\mathrm{selective}} \mid \{\mathcal{E}_{\boldsymbol{Y}} = \mathcal{E}_{\boldsymbol{y}}, \mathcal{Q}(\boldsymbol{Y}) = \mathcal{Q}(\boldsymbol{y})\} \sim \mathrm{Unif}(0, 1),$$

which leads to

$$\mathbb{P}_{\mathrm{H}_0}\left(p_{\mathrm{selective}} \leq \alpha \mid \mathcal{E}_{\boldsymbol{Y}} = \mathcal{E}_{\boldsymbol{y}}, \mathcal{Q}(\boldsymbol{Y}) = \mathcal{Q}(\boldsymbol{y})\right) = \alpha, \, \forall \alpha \in (0, 1).$$

Next, for any $\alpha \in (0, 1)$, we have

$$\mathbb{P}_{\mathrm{H}_0}\left(p_{\mathrm{selective}} \leq \alpha \mid \mathcal{E}_{\boldsymbol{Y}} = \mathcal{E}_{\boldsymbol{y}}\right)$$
$$= \int \mathbb{P}_{\mathrm{H}_0}\left(p_{\mathrm{selective}} \leq \alpha \mid \mathcal{E}_{\boldsymbol{Y}} = \mathcal{E}_{\boldsymbol{y}}, \mathcal{Q}(\boldsymbol{Y}) = \mathcal{Q}(\boldsymbol{Y})\right) \mathbb{P}_{\mathrm{H}_0}\left(\mathcal{Q}(\boldsymbol{Y}) = \mathcal{Q}(\boldsymbol{Y}) \mid \mathcal{E}_{\boldsymbol{Y}} = \mathcal{E}_{\boldsymbol{y}}\right) d\mathcal{Q}(\boldsymbol{y})$$
$$= \alpha \int \mathbb{P}_{\mathrm{H}_0}(\mathcal{Q}(\boldsymbol{Y}) = \mathcal{Q}(\boldsymbol{y}) \mid \mathcal{E}_{\boldsymbol{Y}} = \mathcal{E}_{\boldsymbol{y}}) d\mathcal{Q}(\boldsymbol{y})$$
$$= \alpha.$$

Therefore, we obtain the result in Theorem 4.2 as follows:

$$\mathbb{P}_{\mathrm{H}_0}\left(p_{\mathrm{selective}} \leq \alpha\right) = \sum_{\mathcal{E}_{\boldsymbol{y}}} \mathbb{P}_{\mathrm{H}_0}\left(p_{\mathrm{selective}} \leq \alpha \mid \mathcal{E}_{\boldsymbol{Y}} = \mathcal{E}_{\boldsymbol{y}}\right) \mathbb{P}_{\mathrm{H}_0}(\mathcal{E}_{\boldsymbol{Y}} = \mathcal{E}_{\boldsymbol{y}})$$
$$= \alpha \sum_{\mathcal{E}_{\boldsymbol{y}}} \mathbb{P}_{\mathrm{H}_0}(\mathcal{E}_{\boldsymbol{Y}} = \mathcal{E}_{\boldsymbol{y}})$$
$$= \alpha.$$

# B Selection Event Characterization

In this section, we characterize the selection events $\mathcal{E}_{\boldsymbol{Y}} = \mathcal{E}_{\boldsymbol{y}}$ of deep $k$NN-based anomaly detection (AD). The selection event of deep $k$NN-based AD consists of two components: the selection event related to the $k$NN-based AD, and the selection event related to the deep learning models that perform the transformation into latent features. The former is described in Appendix B.1, and the latter in Appendix B.2. Finally, in Appendix B.3, we describe how to identify the data space that satisfies the selection event and how to compute the selective $p$-values.

## B.1 Selection Event for $k$NN Anomaly Detection

In the selection events of $k$NN-AD, it is necessary to consider events such as selecting the $k$ nearest instances, the anomaly score exceeding a threshold, and determining $k$ based on the data. In the following, we describe these events one by one. It is worth noting that all the events described below can be collectively represented by a set of linear inequalities, which facilitates the computation of truncation regions for the truncated normal distribution used in selective $p$-value calculations.

**Selection event for $k^{\text{th}}$ nearest neighbor**  The anomaly detection decision based on Eq. (3) and the test statistic of Eq. (7) depend on the selection of the $k^{\text{th}}$ nearest neighbor instance of the test instance $\boldsymbol{X}^{\text{test}}$. Therefore, the condition on the $k^{\text{th}}$ nearest neighbor instance is required. Specifically, by conditioning on

$$\text{dist}_\phi(\boldsymbol{X}^{\text{test}}, \boldsymbol{X}_{o(k)}) \geq \text{dist}_\phi(\boldsymbol{X}^{\text{test}}, \boldsymbol{X}_{o(k')}) \tag{18}$$

for $k' = 1, \ldots, k-1$, and

$$\text{dist}_\phi(\boldsymbol{X}^{\text{test}}, \boldsymbol{X}_{o(k)}) \leq \text{dist}_\phi(\boldsymbol{X}^{\text{test}}, \boldsymbol{X}_{o(k')}) \tag{19}$$

for $k' = k+1, \ldots, n$, we can consider only cases where the $k$-the nearest neighbor is the same as the observed case. Hereafter, the conditions in Eq.(18) and Eq.(19) are collectively represented as $\mathcal{N}_{\boldsymbol{Y}} = \mathcal{N}_{\boldsymbol{y}}$.

**Selection Events for $k$-Medoids**  The test statistic in Eq. (7) also depends on the selection of the $k$-medoid instance. Therefore, conditions on the $k$-medoid instance are also necessary. Specifically, using Eq. (5), it is expressed as follows:

$$\sum_{i \in \mathcal{K}/\{\text{med(k)}\}} \text{dist}_\phi(\boldsymbol{X}_i, \boldsymbol{X}_{\text{med(k)}}) \leq \sum_{i \in \mathcal{K}/\{j\}} \text{dist}_\phi(\boldsymbol{X}_i, \boldsymbol{X}_j) \quad j \in \mathcal{K}, j \neq \text{med(k)}. \tag{20}$$

Hereafter, the condition in equation (20) is expressed as $\mathcal{M}_{\boldsymbol{Y}} = \mathcal{M}_{\boldsymbol{y}}$.

**Selection event for anomaly score**  Since the statistical test is performed only on test instances selected in the AD, it is essential to consider the selection events associated with it. A test instance is selected and if its anomaly score, as defined in Eq. (3), exceeds a threshold $\theta$. The condition for the anomaly score is written as

$$\log \text{dist}_\phi \left( \boldsymbol{X}^{\text{test}}, \boldsymbol{X}_{o(k)} \right) - \frac{\log k}{d} \geq \theta. \tag{21}$$

With the conditions in Eq.(21), we can characterize the selection event that the test case $\boldsymbol{X}^{\text{test}}$ is selected in AD. Hereafter, the condition in Eq.(21) is represented as $\mathcal{K}_{\boldsymbol{Y}} = \mathcal{K}_{\boldsymbol{y}}$.

**Selection event for data-driven selection of $k$**  In the case of the data-driven option for determining the number of neighbors $k$, its effect must also be appropriately considered as a selection event. For example, consider the scenario where $k_1, \ldots, k_K$ are candidate values for $k$, and the candidate that maximizes the anomaly score in Eq. (3) is selected. Let the selected $k \in \{k_1, \ldots, k_K\}$ be denoted as $k^*$. Then, the selection event is simply given by $\log \text{dist}(\boldsymbol{x}^{\text{test}}, \boldsymbol{x}_{o(k^*)}) - \frac{\log k^*}{d} \geq \log \text{dist}(\boldsymbol{x}^{\text{test}}, \boldsymbol{x}_{o(k_t)}) - \frac{\log k_t}{d}, \forall t \in [K]$. In the case of data-driven option to determine $k$, in addition to the four selection events mentioned above, this event must also be incorporated as an additional condition. Hereafter, we denote this selection event as $\mathcal{S}_{\boldsymbol{Y}} = \mathcal{S}_{\boldsymbol{y}}$.

## B.2 Selection Event for Deep Learning Models

When using $k$-nearest neighbors AD with feature representations from a pre-trained deep learning model, the influence of the model should be considered as a selection event. SI for deep learning has been discussed in prior studies, and tools like the software facilitate the analysis of selection events in these models. In this study, we employ methods from earlier research to calculate selective $p$-values, taking into account selection events related to deep learning models. The basic idea in these methods involves decomposing the model into components and representing each as a piecewise linear function. For example, operations in a CNN such as convolution, ReLU activation, max pooling, and up-sampling are represented as piecewise linear functions. In the experiment, we utilize the feature representation of a CNN model pre-trained on the ImageNet database. This model is represented precisely as a composition of piecewise linear functions. We explain the selection events regarding the deep learning model that transforms an image instance $\boldsymbol{x}_i \in \mathbb{R}^d$ to a latent feature vector $\boldsymbol{z}_i \in \mathbb{R}^{\tilde{d}}$. We consider a deep learning model that consists of sequential piecewise-linear functions (e.g., convolution, ReLU activation, max pooling, and up-sampling). Obviously, the composite function of those piecewise-linear functions maintains its piecewise-linear nature. Thus, within a specific real space in $\mathbb{R}^d$, the deep learning model simplifies to a linear function, which can be expressed as:

$$\phi_{\mathrm{DL}}(\boldsymbol{x}_i) = \boldsymbol{B} + \boldsymbol{W}\boldsymbol{x}_i \quad \text{if } \boldsymbol{x}_i \in \mathcal{P}, \tag{22}$$

where $\boldsymbol{B} \in \mathbb{R}^{\tilde{d}}$ and $\boldsymbol{W} \in \mathbb{R}^{\tilde{d} \times d}$ represent the bias and weight matrices, and $\mathcal{P} \subseteq \mathbb{R}^d$ is a polytope where $\phi_{\mathrm{DL}}$ acts as a linear function. The polytope can be characterized by a set of linear inequalities. For details on computing these linear inequalities, see [53]. Let us denote the set of polytopes for all instances in $\boldsymbol{Y}$ as:

$$\mathcal{D}_{\boldsymbol{Y}} := \{\mathcal{P} \mid \boldsymbol{X}_i \in \boldsymbol{Y}, \boldsymbol{X}_i \in \mathcal{P}\}. \tag{23}$$

Hereafter, we denote the selection event as $\mathcal{D}_{\boldsymbol{Y}} = \mathcal{D}_{\boldsymbol{y}}$.

## B.3 Computing Selective $p$-values

Based on the discussions in Appendix B.1 and B.2, selective $p$-values in (14) can be rewritten as follows:

$$p_{\text{selective}} := \mathbb{P}_{\mathrm{H}_0}\left(T(\boldsymbol{Y}) \geq T(\boldsymbol{y})|\mathcal{N}_{\boldsymbol{Y}} = \mathcal{N}_{\boldsymbol{y}}, \mathcal{M}_{\boldsymbol{Y}} = \mathcal{M}_{\boldsymbol{y}}, \mathcal{K}_{\boldsymbol{Y}} = \mathcal{K}_{\boldsymbol{y}}, \mathcal{S}_{\boldsymbol{Y}} = \mathcal{S}_{\boldsymbol{y}}, \mathcal{D}_{\boldsymbol{Y}} = \mathcal{D}_{\boldsymbol{y}}, \mathcal{Q}_{\boldsymbol{Y}} = \mathcal{Q}_{\boldsymbol{y}}\right). \tag{24}$$

Although the calculation of this selective $p$-values is complex, the marginalization of $\mathcal{N}_{\boldsymbol{Y}} = \mathcal{N}_{\boldsymbol{y}}, \mathcal{D}_{\boldsymbol{Y}} = \mathcal{D}_{\boldsymbol{y}}, \mathcal{S}_{\boldsymbol{Y}} = \mathcal{S}_{\boldsymbol{y}}$ is effectively utilized and existing SI research methods. We specifically use the parametric programming (pp)-based method from previous studies [38]. In SI, statistical inference is based on the probability measure within the subspace $\mathcal{Z}$ of the data space $\mathbb{R}^{(1+n)d}$ where selection event conditions are met. By conditioning on the selection event for the nuisance component, $\mathcal{Q}_{\boldsymbol{Y}} = \mathcal{Q}_{\boldsymbol{y}}$, $\mathcal{Z}$ reduces to a one-dimensional subspace (see Theorem 4.1 and its proof in Appendix A.1). The selection events are formulated as unions of intersections of linear or quadratic inequalities, suitable when using $L_1$ or $L_2$ distances for $k$-nearest neighbors. $\mathcal{Z}$ consists of finite number of intervals along a line in the $(1 + n)d$-dimensional space, and the pp-based method systematically enumerates all intervals that meet these conditions.

Since the noise is Gaussian, the test statistic $T(\boldsymbol{Y})$ under the null hypothesis $\mathrm{H}_0$ follows a one-dimensional truncated Gaussian distribution within the subspace $\mathcal{Z}$, comprising finite intervals along a line. The selective $p$-value is calculated as the tail probability of this truncated distribution. Early SI research often simplified calculations by assuming $\mathcal{Z}$ as a single interval under additional conditions, which still controls the false detection probability but reduces detection power. In our problem, a similar simplification can be considered by enforcing $\mathcal{Z}$ to be a single interval. In the experiments in §5, we conduct an ablation study comparing this simple approach (denoted as `w/o-pp`) as one of the baselines.

# C Details of the Experiments

## C.1 Details of Synthetic Data Generation

This subection provides additional details regarding the generation of synthetic datasets used in Section 5.2. We describe both the parametric and semi-parametric settings. To illustrate the two

data-generation settings, we present in Figure 6 the distributions of the training samples in a two-dimensional example ($d = 2$). In the parametric setting, all samples are centered around the origin. In contrast, in the semi-parametric setting, the samples are distributed around different mean vectors $\boldsymbol{s}_i$, producing a mixture of Gaussian clusters. This visualization clarifies the structural difference between the two settings and the increased heterogeneity in the semi-parametric case.

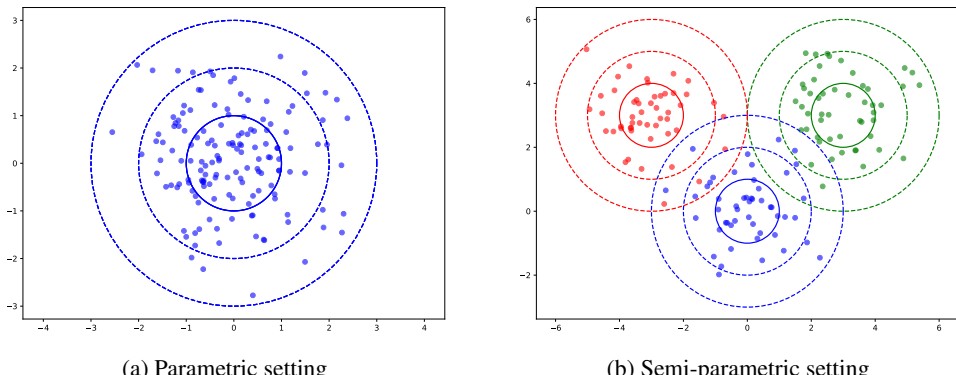

(a) Parametric setting        (b) Semi-parametric setting

Figure 6: Visualization of the data-generation process in the parametric and semi-parametric settings for $d = 2$. In the parametric case, all training samples are drawn from a single Gaussian distribution centered at the origin. In the semi-parametric case, each training sample is drawn from a Gaussian distribution with a randomly shifted mean vector $\boldsymbol{s}_i$, resulting in a heterogeneous distribution.

## C.2 Additional Type I Error Rate Results

We also conducted experiments to investigate the Type I error rate when the data dimension $n$, $d$ and the number of neighbors $k$ were varied in the parametric and semi-parametric setting. Specifically, we varied $d \in \{5, 10, 15, 20\}$, $k \in \{1, 2, 5, 10\}$ and $n \in \{100, 200, 500, 1000\}$, while setting the default parameters as $d = 5$, $k = 3$ and $n = 100$. In all cases, we generated the datasets in the same way as in the experiments on synthetic datasets (§5.2). The results are shown in Figures 7, 8 and 9.

To further assess the robustness of our method, we conducted experiments on datasets that deviate from the normal distribution. Specifically, data are sampled from the exponentially modified Gaussian (EMG), generalized normal distribution (GND), skew normal distribution (SND), and Student's $t$-distribution. The degree of deviation from the normal distribution is quantified using the Wasserstein distance $l$, and we evaluate the Type I error rate for each case by varying $l \in \{0.01, 0.02, 0.03, 0.04\}$. The results are shown in Figure 10.

## C.3 Additional Power Results

We also conducted experiments to investigate the power when the number of training data $n$, the data dimension $d$ and the number of neighbors $k$ are varied in the parametric and semi-parametric setting. We varied $n \in \{100, 200, 500, 1000\}$, $d \in \{5, 10, 15, 20\}$ and $k \in \{1, 2, 5, 10\}$ while setting the default parameters as $n = 100$, $d = 5$, $k = 3$ and signal strength $\delta = 5$. Furthermore, we conducted additional experiments where $n$ and $d$ was varied, considering the case where $k$ was adaptively selected from $\in \{1, 2, 5, 10\}$ in a data-driven manner. In all cases, we generated the datasets in the same way as in the experiments on synthetic datasets (§5.2). The results are shown in Figures 11, 12, 13, and 14.

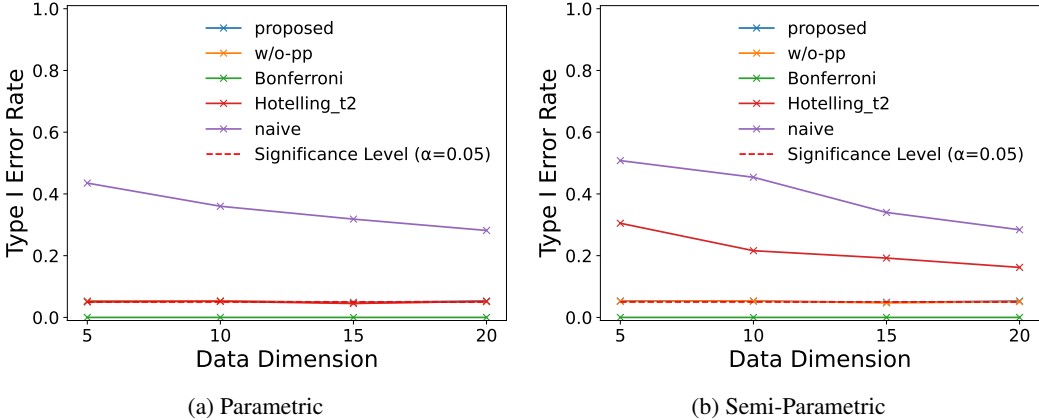

(a) Parametric

(b) Semi-Parametric

Figure 7: Results of Type I error rate when varying the date dimension $d$. `proposed`, `w/o-pp`, and `Bonferroni` successfully control the Type I error rate across all settings. `naive` fails and the results of `Bonferroni` are almost zero, because it is too conservative. Since `Hotelling_t2` does not involve a parameter $k$, its value remains unchanged. `Hotelling_t2` also fails in the semi-parametric setting.

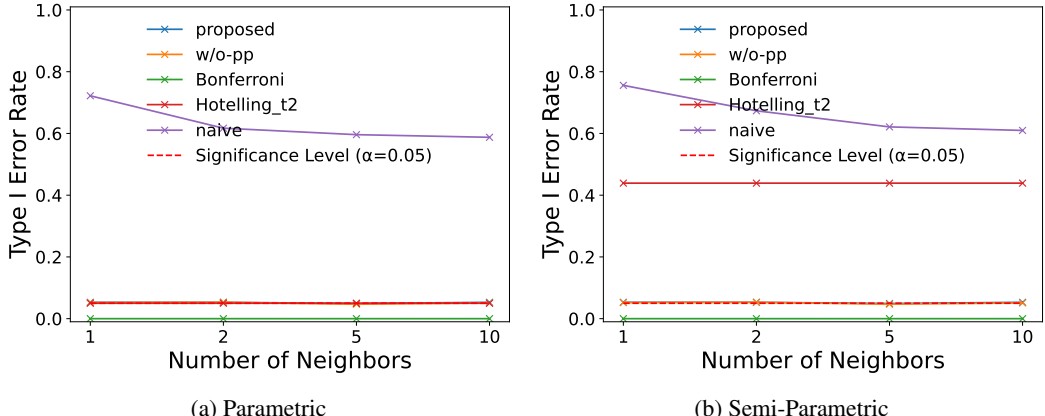

(a) Parametric

(b) Semi-Parametric

Figure 8: Results of Type I error rate when varying the number of neighbors $k$. `proposed`, `w/o-pp`, and `Bonferroni` successfully control the Type I error rate across all settings. `naive` fails and the results of `Bonferroni` are almost zero, because it is too conservative. Since `Hotelling_t2` does not involve a parameter $k$, its value remains unchanged in the both settings. In the semi-parametric setting, `Hotelling_t2` fails to control the Type I error rate.

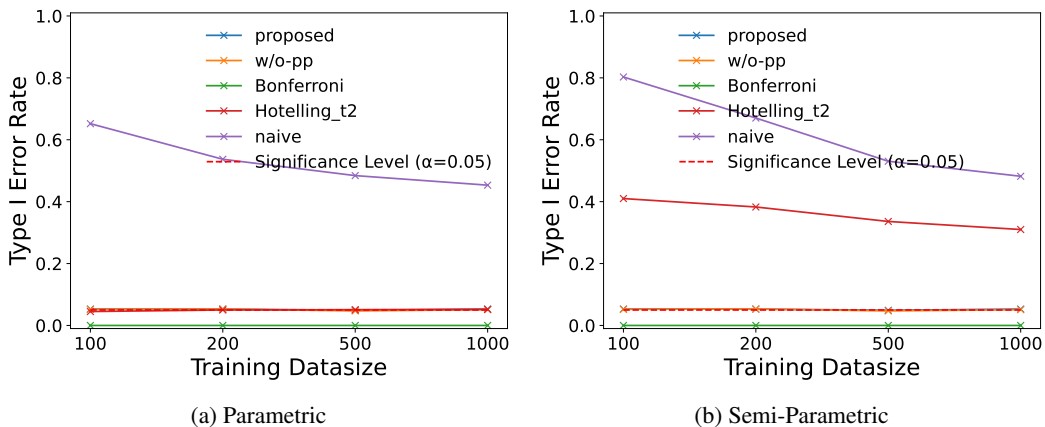

(a) Parametric

(b) Semi-Parametric

Figure 9: Results of Type I error rate when varying the number of datasize $n$. `proposed`, `w/o-pp`, and `Bonferroni` successfully control the Type I error rate across all settings. `naive` fails and the results of `Bonferroni` are almost zero, because it is too conservative. `Hotelling_t2` also fails in the semi-parametric setting.

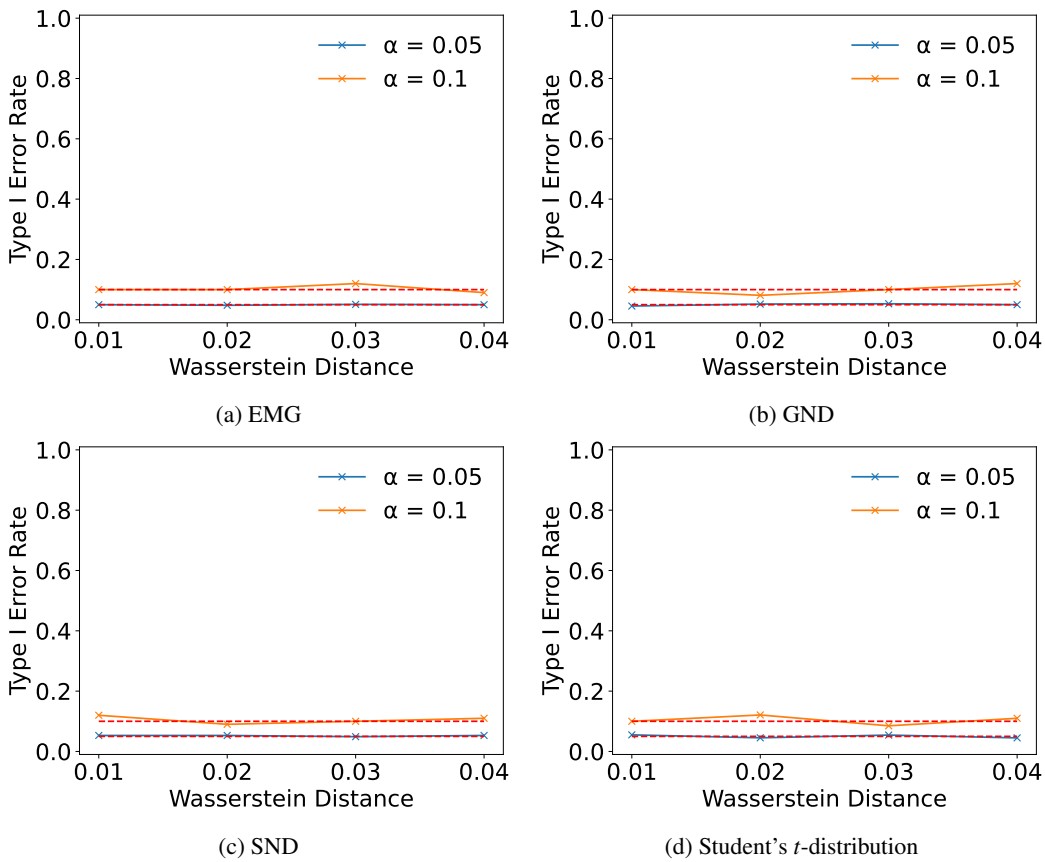

(a) EMG

(b) GND

(c) SND

(d) Student's $t$-distribution

Figure 10: Results of Type I error rate when varying the Wasserstein distance $l$. `proposed` successfully control the Type I error rate in both significance levels $\alpha\{0.05, 0.1\}$.

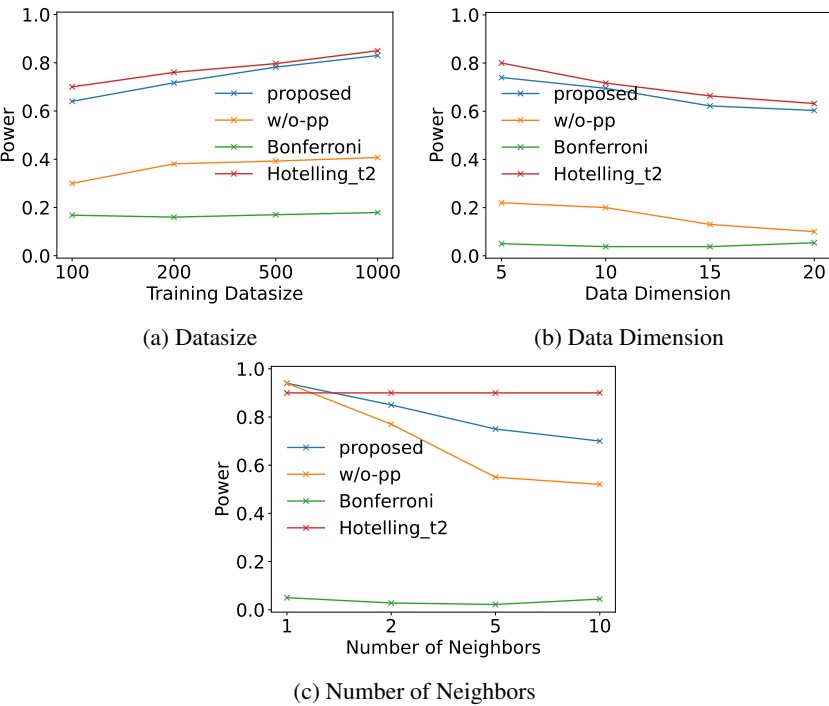

(a) Datasize          (b) Data Dimension

(c) Number of Neighbors

Figure 11: These are in the parametric setting. Power for a fixed number of neighbors $k$ . The results show the effect of varying the training dataset size $n$, the data dimension $d$, and $k$. Our proposed method (`proposed`) and `Hotelling_t2` outperformed other methods across all settings.

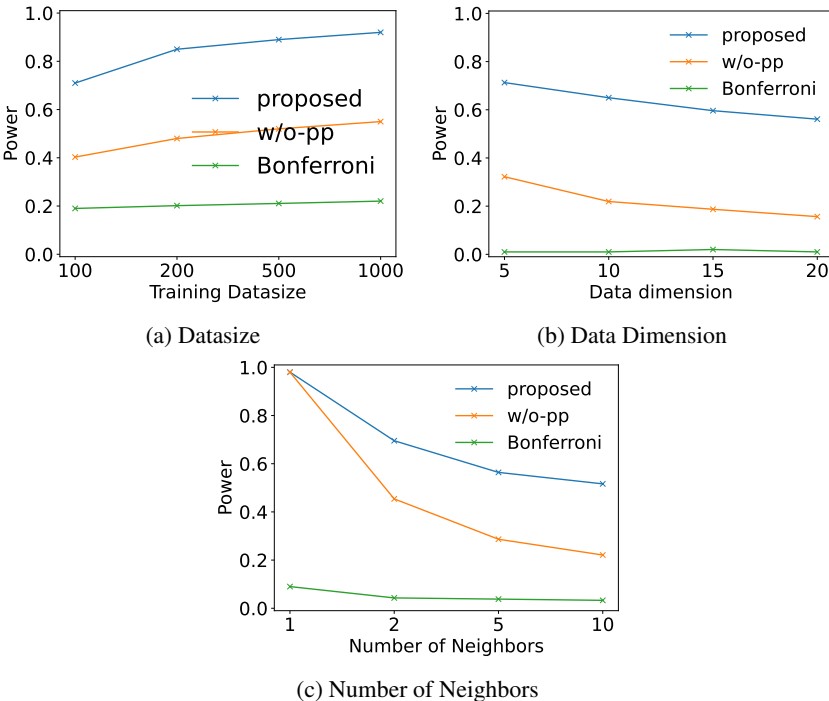

(a) Datasize          (b) Data Dimension

(c) Number of Neighbors

Figure 12: These are in the semi-parametric setting. Power for a fixed number of neighbors $k$. The results show the effect of varying the training dataset size $n$, the data dimension $d$, and $k$. Our proposed method (`proposed`) outperformed other methods across all settings.

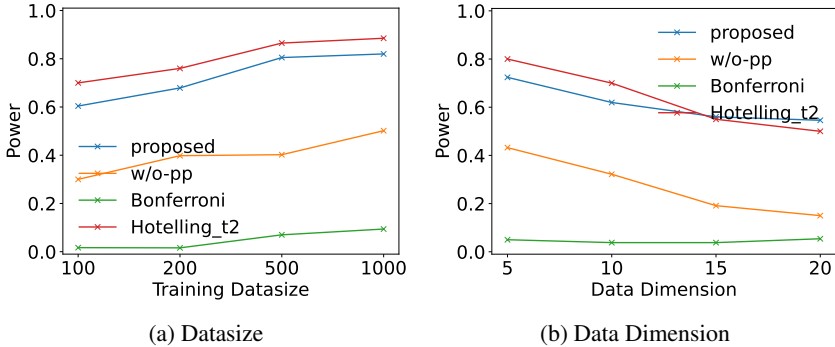

(a) Datasize         (b) Data Dimension

Figure 13: These are in the parametric setting. Power for an adaptively selected number of neighbors $k$. The results show the effect of varying the training dataset size $n$ and the data dimension $d$. Our proposed method (`proposed`) and `Hotelling_t2` outperformed other methods across all settings.

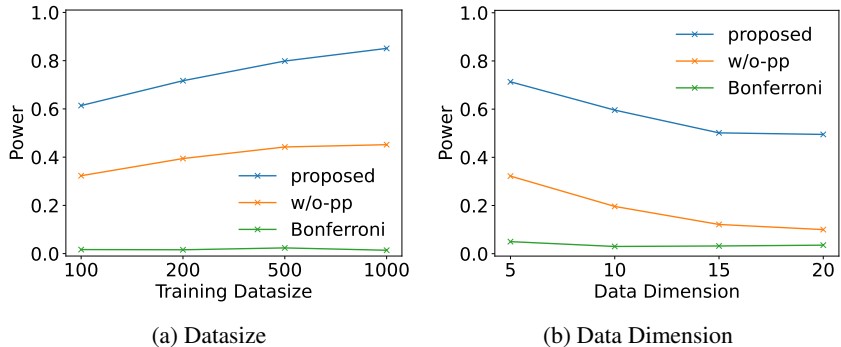

(a) Datasize         (b) Data Dimension

Figure 14: These are in the semi-parametric setting. Power for an adaptively selected number of neighbors $k$. The results show the effect of varying the training dataset size $n$ and the data dimension $d$. Our proposed method (`proposed`) outperformed other methods across all settings.

## C.4 Details of Tabular Datasets

We used the following 10 real datasets from the Kaggle Repository. All datasets are licensed under the CC BY 4.0 license.

- *Heart*: Dataset for predicting heart attacks
- *Money*: Dataset on financial transactions in a virtual environment
- *Fire*: Dataset on fires in the MUGLA region in June
- *Cancer*: Dataset related to breast cancer diagnosis
- *Credit*: Dataset on credit card transactions
- *Student*: Dataset related to student performance
- *Bankruptcy*: Dataset on company bankruptcies
- *Drink*: Dataset on the quality of drinking water
- *Nuclear*: Dataset on pressurized nuclear reactors
- *Network*: Dataset on anomaly detection in virtual network environments

## C.5  Experimental Results on Image Data Examples

We evaluated `proposed` and `naive` on the 10 datasets from MVTec AD dataset. The datasets used in this study are *Carpet*, *Grid*, *Leather*, *Tile*, *Wood*, *Bottle*, *Capsule*, *Metal Nut*, *Transistor*, and *Zipper*. Examples except for those shown in the Figure 5 from each dataset are shown in Figure 15. In each example, we present patches corresponding to true negative and true positive cases, along with both the naive $p$-value and the selective $p$-value.

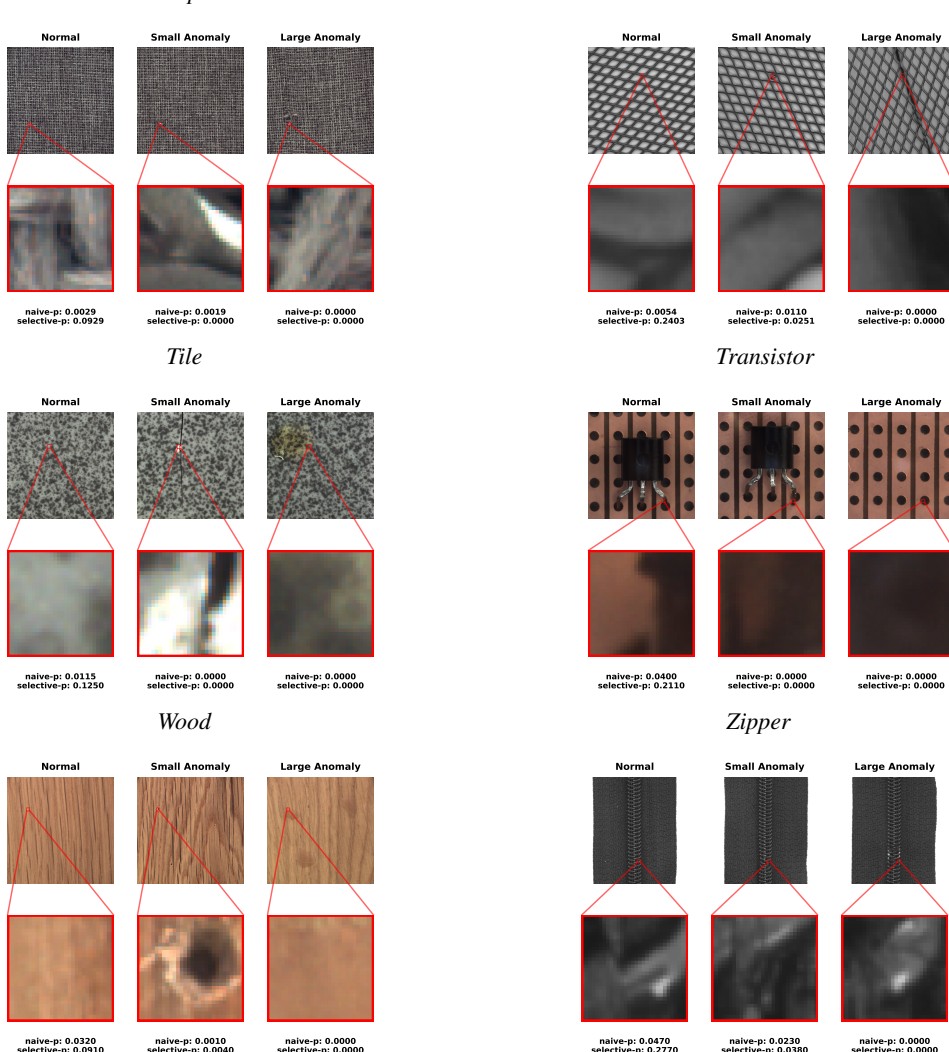

Figure 15: Experimental results of 6 datasets from MVTec AD dataset. For each dataset, one normal example (left) and two anomaly examples (center, right) are showed. For each example, the top row displays the original image used for testing along with the patch location (marked in red), while the bottom row presents the extracted patch image. For all normal examples, the naive $p$-value is below the significance level $\alpha = 0.05$ (false positive), whereas the proposed selective $p$-value correctly results in a true negative. For all anomaly examples, the selective $p$-value successfully detects anomalies.

