# OpenReview forum: "Quantifying Statistical Significance of Deep Nearest Neighbor Anomaly Detection via Selective Inference"
_NeurIPS.cc/2025/Conference — NeurIPS 2025 poster_

### Official Review · Reviewer_VcWS · 2025-06-09

**Clarity:** 3
**Significance:** 3
**Originality:** 3
**Rating:** 5
**Confidence:** 2

**Summary:**

This paper proposes a framework to quantify the statistical significant of anomalies detected through deep kNN based anomaly detection in the form of p-values for hypothesis testing via Selective Inference (SI)

**Questions:**

1. Is this the first paper to consider SI for AD? Or only for deep kNN-based AD? If so, how does this methodology differ from others proposed in other settings?

2. How does this result translate to other relevant tasks such as OOD detection?

**Ethical Concerns:**

["NO or VERY MINOR ethics concerns only"]

**Final Justification:**

Technically strong and novel contribution and the authors responses are convincing.

**Limitations:**

No discussion of negative societal impact, though model limitations are discussed.

**Quality:**

4

**Strengths And Weaknesses:**

The paper is thoughtfully written and easy to understand.

Experiments are comprehensive, covering a variety of data types (tabular, image) and a range of datasets.

It is difficult to find fault with this paper, though I would say it would be beneficial to make more effort explaining the details of other SI methods to provide clearer grounding of this work within the context of other literature. Also, Figure 4 is not particularly easy to comprehend. The authors may consider alternative ways to plot their results.

---

> ### Author Rebuttal · Authors · 2025-07-31
>
> We thank the reviewer for your feedback.
>
> >(Weakness-1-1) I would say it would be beneficial to make more effort explaining the details of other SI methods to provide clearer grounding of this work within the context of other literature.
>
> In the camera-ready version, we plan to expand the related works section to include more detailed discussion of other SI methods.
>
> > (Weakness-1-2) Also, Figure 4 is not particularly easy to comprehend. The authors may consider alternative ways to plot their results.
>
> Additionally, we will revise the visualization and caption of Fig. 4 to make the differences between our method and the baseline methods more easily understandable.
>
> >(Question-1). Is this the first paper to consider SI for AD? Or only for deep kNN-based AD? If so, how does this methodology differ from others proposed in other settings?
>
> To the best of our knowledge, this is the first study to apply SI to anomaly detection. While SI has been extensively studied in the statistical literature, its application to machine learning methods remains very limited. One related work is the application of SI to change-point detection in time series data [1]. However, that line of research focuses on testing the significance of global changes in the entire sequence, which differs from our setting where the goal is to assess the abnormality of individual data points. Another related work is the application of SI to robust regression [2], where inference is performed on regression coefficients obtained after excluding outliers. However, the statistical significance of the detected outliers themselves is not addressed. Moreover, to the best of our knowledge, our work is also the first to apply SI in the context of data analysis performed in a latent space induced by deep learning models. We plan to expand the discussion of these points in the related works section of the camera-ready version.
>
> [1] https://arxiv.org/pdf/1606.03552
> [2] https://arxiv.org/abs/1711.10635
>
> >(Question-2). How does this result translate to other relevant tasks such as OOD detection?
>
> We appreciate the suggestion highlighting this interesting research direction. While it would be necessary to formulate the corresponding selection event (e.g., $\varepsilon_{\boldsymbol{Y}} = \varepsilon_{\boldsymbol{y}}$ as in Eq (6)), we believe that our framework is in principle applicable to other tasks such as OOD. For example, OOD detection based on Mahalanobis distance in the latent space [3] shares structural similarities with our setting, and thus may allow for a natural extension of our method.
>
> [3] https://arxiv.org/pdf/2505.18032

---

> > ### Comment · Reviewer_VcWS · 2025-08-07
> >
> > Thank you for the response. I maintain my score.

---

### Official Review · Reviewer_Awbd · 2025-06-28

**Clarity:** 2
**Significance:** 2
**Originality:** 2
**Rating:** 4
**Confidence:** 4

**Summary:**

This paper applies selective inference techniques to KNN anomaly detection, aiming to provide statistical guarantees for the detection process. The authors present a methodological framework, prove supporting theorems, and evaluate the approach on synthetic and real-world datasets.

**Questions:**

Please see the weakness part.

**Ethical Concerns:**

["NO or VERY MINOR ethics concerns only"]

**Final Justification:**

The rebuttal addressed my concern.

**Limitations:**

yes

**Paper Formatting Concerns:**

no clear issues.

**Quality:**

2

**Strengths And Weaknesses:**

Strengths:

The paper clearly articulates its core contribution: establishing a selective inference framework tailored for kNN-based anomaly detection, supported by theoretical results and empirical validation. The inclusion of asymptotic results and experiments helps bridge a gap in the statistical understanding of kNN anomaly detection methods.

Weaknesses:

The theoretical contributions appear limited in novelty. The paper primarily establishes asymptotic distributions under standard Gaussian assumptions, following a relatively well-established framework. It does not clearly demonstrate the unique advantage of the proposed selective inference method theoretically over alternatives such as empirical p-value–based methods or bonferroni correction or multiple testing techniques like FDR.

The Gaussian assumptions throughout the theory are too strong. This restricts the generality of the results and may undermine applicability to real-world anomaly detection settings, where sub-Gaussian, sub-exponential, or heavy-tailed distributions are common. This issue also extends to the synthetic experiments, which are very limited to parametric and semi-parametric settings under Gaussian assumptions.

Moreover, the paper mentions general applicability of the SI framework across learning methods, yet does not sufficiently articulate what makes this adaptation to kNN unique. This raises questions about the novelty and specificity of the proposed approach.

Certain components, such as the anomaly score in Equation (1), appear to be adopted from existing literature without strong motivation or justification. Similarly, the test statistic used is tightly coupled to the Gaussian assumption without any data-driven adjustment, further weakening its practical utility.

---

> ### Author Rebuttal · Authors · 2025-07-31
>
> We thank the reviewer for your feedback.
>
> >(Weakness-1) The theoretical contributions appear limited in novelty. The paper primarily establishes asymptotic distributions under standard Gaussian assumptions, following a relatively well-established framework. It does not clearly demonstrate the unique advantage of the proposed selective inference method theoretically over alternatives such as empirical p-value–based methods or bonferroni correction or multiple testing techniques like FDR.
>
> First of all, we emphasize that Theorem 4.2 provides a finite-sample guarantee of Type I error control, which distinguishes our framework from many existing asymptotic approaches. Our Gaussian assumptions are also not overly restrictive (please refer to our response to Weakness-2 for further details). Regarding comparisons with alternative methods, we experimentally demonstrate that our approach achieves higher statistical power than Bonferroni correction, which is known to be overly conservative and often results in reduced sensitivity (see Figures 3 and 4). Unlike Bonferroni, which lowers the Type I error rate at the expense of power, our selective inference framework controls the Type I error rate exactly at the user-specified level, as guaranteed by Theorem 4.2. We selected Bonferroni as a baseline because it is applicable when the number of hypotheses is known. In contrast, other multiple testing procedures, such as Holm's method, require computing nominal p-values for all possible hypotheses, which is an intractable task due to the combinatorial explosion. Furthermore, since we aim to determine whether a single test instance is anomalous, permutation-based tests are not applicable. We will clarify these points in the revised manuscript.
>
> > (Weakness-2) The Gaussian assumptions throughout the theory are too strong.
>
> We would like to clarify that our framework does not assume the overall distribution of normal instances to be Gaussian. As described in Eq.(2), our setting is semi-parametric in the sense that we make no distributional assumptions on the signal component, which can follow an arbitrary and potentially complex distribution. The Gaussian assumption is imposed only on the additive noise component. Unlike fully parametric tests such as Hotelling's $T^2$, which assume the entire observation is Gaussian, our assumption is significantly less restrictive. Furthermore, to assess the robustness of our method under noise model misspecification, we provide additional experiments in Appendix C.2 (Figure 8), where the noise component follows non-Gaussian distributions. These experimental results indicate that our method remains effective and continues to control the Type I error rate even when the Gaussian noise assumption is violated.
>
> > (Weakness-3) Moreover, the paper mentions general applicability of the SI framework across learning methods, yet does not sufficiently articulate what makes this adaptation to kNN unique. This raises questions about the novelty and specificity of the proposed approach.
>
> Our main contribution is to introduce a statistically principled framework for deep learning–based anomaly detection that explicitly controls the trade-off between false positives and true positives. To the best of our knowledge, no existing work provides such a general and reliable framework in this context. We chose the deep k-NN method as a proof-of-concept because it is simple, widely used, and allows us to clearly illustrate our framework's utility. While our selective inference approach is general in principle, adapting it to other anomaly detection methods requires characterizing the selection event specific to each algorithm (i.e., $\varepsilon\_{\boldsymbol{Y}} = \varepsilon\_{\boldsymbol{y}}$ in Eq (6)), which involves nontrivial engineering effort. Building a systematic pipeline for applying our framework across broader classes of deep anomaly detectors is an important direction for future work.
>
> >(Weakness-4) Certain components, such as the anomaly score in Equation (1), appear to be adopted from existing literature without strong motivation or justification.
>
> The anomaly score used in our study is based on a classical definition. Specifically, it corresponds to the log-transformed version of Eq (3.2) in [1] up to constant terms. We will explicitly clarify this point in the camera-ready version.
>
> [1] Loftsgaarden and Quesenberry. A Nonparametric Estimate of a Multivariate Density Function. Ann. Math. Statist. 36(3): 1049-1051 (1965)
>
> > (Weakness5) Similarly, the test statistic used is tightly coupled to the Gaussian assumption without any data-driven adjustment, further weakening its practical utility.
>
> We are afraid that there may be a misunderstanding regarding the role of the Gaussian assumption and the nature of our test statistic. As mentioned in our response to Weakness-2, our method does not assume Gaussianity for the data distribution (see Eq.(2)). While the noise model used in calculating the sampling distribution of the test statistic assumes Gaussianity (a common and analytically tractable choice), the test statistic itself is not specialized for Gaussian data. Moreover, contrary to the reviewer's comment, our test statistic is in fact data-driven. Specifically, $T(\boldsymbol{Y}) = \\|\boldsymbol{\eta}\_{\boldsymbol{y}}^t \boldsymbol{Y}\\|_2^2$ is constructed using the vector $\boldsymbol{\eta}\_{\boldsymbol{y}}$, which depends on the deep k-NN structure conditioned on the observed data point $\boldsymbol{y}$. This data-dependent construction is central to our methodological contribution, which enables selective inference for deep data-driven hypotheses. Within the selective inference framework, we condition on the selection event associated with this $\boldsymbol{\eta}\_{\boldsymbol{y}}$, making the sampling distribution of the data-dependent statistic tractable.

---

> > ### Comment · Reviewer_Awbd · 2025-08-05
> > **Response**
> >
> > Thanks for the reviewer for giving the detailed response. I will raise my score.

---

### Official Review · Reviewer_ZerL · 2025-06-30

**Clarity:** 3
**Significance:** 2
**Originality:** 3
**Rating:** 5
**Confidence:** 3

**Summary:**

This work introduces a selective inference framework for quantifying the statistical significance of the a deep kNN anomaly detector. The proposed method studies what would the p-value of a test observation be if we assume that the signal component (which is modelled non-parametric) is the same as the k-nearest neighbour. Nonetheless, since the computation of this p-value is intractable because it'll mean to model a complex neural network, the authors use Selective Inference to achieve a tractable solution (by conditioning the significance test). Numerical examples are performed considering synthetic, tabular and image data, showing that the selective inference provides a good measure of statistical significance while having the greatest power among all of the other baseline methods. Overall, this work is an applied work about selective inference for the anomaly distribution task.

**Questions:**

1. Could you please elaborate on how you reached to $\mathbf{Y} = \mathcal{U}(\mathbf{y}) + \lVert P\mathbf{Y}  \rVert_2^2 \mathcal{V}(\mathbf{Y})$ andh why it is not actually $\mathbf{Y} = \mathcal{U}(\mathbf{y}) + \lVert P\mathbf{Y}  \rVert \mathcal{V}(\mathbf{Y})$ (as per above)? If indeed you made a mistake, how does that would affect your analysis? (If I'm making a mistake please ignore this question).

2. How could this improve upon Anomaly Detection in industrial settings?

3. Could you provide more results for a different image dataset? The MVTec is already solved.

4. How would this analysis translate to other Anomaly Detection methods? Such as Teacher-Student models?

5. Isn't Eq (4) the same as Eq (5)? But you say Eq (4) is intractable but Eq (5) can be computed. I might be wrong, but could you clarify this?

I'd be happy to change my score if these questions are answered satisfactory, but I'm also going to consider other factors such as discussion with other reviewers and the main weaknesses I found.

**Ethical Concerns:**

["NO or VERY MINOR ethics concerns only"]

**Final Justification:**

The authors addressed my main concern about the derivation of Theorem 4.1, mainly regarded as a typo, which they claim it does not affect the performance. The authors also compromised to fix all the typos and provide further clarification in the final version of the paper. Therefore, I increase my original rating from kBorderline Reject to Accept.

**Limitations:**

Yes

**Quality:**

2

**Strengths And Weaknesses:**

The work stablishes a good motivation on why using Selective Inference for anomaly detection could give a valid statistical significance measure of a prediction for an anomaly detector. The mathematical framework is mostly well defined and explained, although there are some parts that lack rigorosity. The experiments show that Selective Inference is in fact a good method for getting an accurate p-value. Therefore, I think the work is technically solid and addresses an interesting problem.

Nonetheless, I still found some weaknesses that makes me lean towards a reject. I have three main criticisms: (1) the work is limited to a single method for anomaly detection (deep kNN); (2) it is not clear how this actually helps Anomaly Detection; and (3) I think there might be an error in the proof of Theorem 4.1, which could potentially invalidate some of the analsys . For (1), there are several methods for anomaly detection, and choosing a single method makes me question the significante of the work. Several methods have shown to work as good or better than deep kNN for AD, so it'll be great to discuss how SI could be applied to these methods (if it can). Additionally, the evaluation is carried out in rather limited datasets. I find the synthetic dataset not sufficiently strong to explore the gains of SI in real-life scenarios, and the analysis of a single method in the MVTec dataset is also underwhelming, since this dataset has been practicallty solved using different methods. For (2), I think that while the authors clearly state the motivation, it would be great to discuss how this actually could help in anomaly detection. Regarding (3), in the proof of Theorem 4.1, you have this:

\begin{equation}
(I_{(n+1)d}-P)\mathbf{Y} = \mathcal{U}(\mathbf{y})
\end{equation}

Then we'll have:
\begin{equation}
\mathbf{Y} = \mathcal{U}(\mathbf{y}) + P\mathbf{Y}
\end{equation}

Which would be equivalent to
\begin{equation}
\mathbf{Y} = \mathcal{U}(\mathbf{y}) + \frac{\lVert P\mathbf{Y}  \rVert P\mathbf{Y}}{\lVert P\mathbf{Y} \rVert}
\end{equation}

Which, according to your notation, it'd be:
\begin{equation}
\mathbf{Y} = \mathcal{U}(\mathbf{y}) + \lVert P\mathbf{Y}  \rVert \mathcal{V}(\mathbf{Y})
\end{equation}

But you have
\begin{equation}
\mathbf{Y} = \mathcal{U}(\mathbf{y}) +  \mathcal{V}(\mathbf{Y}) \mathbf{z} =  \mathcal{U}(\mathbf{y}) + \lVert P\mathbf{Y}  \rVert_2^2 \mathcal{V}(\mathbf{Y})
\end{equation}

I think there might be a problem here that probably would invalidate most of the analysis.

Additionally, I also think that the work could be explained better. For instance, adding more detail into where do the condfitions in the equation between lines 197 and 198 come from. I understand that you might refer to a detailed work about SI, and that;s fine, but for the sake of the analysis in this paper I think it is important to discuss it. Is that nuisance parameter the same for all the SI problems? finally, I think some qualitative examples in the main paper would be good.

Other minor weaknesses:
- It'd be a good idea to add quantitative results in the abstract
- L18: Tsehe -> This
- L36: tereby -> thereby, by making...
- L66: It's important to mention why quantifying statistical significance for such AD methods would be important
- L102: I think it's more self-supervised rather than semi-supervised, but I leave it to the author
- L107: $x_i$ -> $\mathbf{x}_i$
- L123-124. Any ref aboyut the heuristic method for choosing k?
- Please add numbering to *all* equations
- Eq between L175 and L176: I think $\boldsymbol{\eta}_y$ depends on $k$ but it's neither   $\boldsymbol{\eta}_k$ or  $\boldsymbol{\eta}_y (k)$. I think there should be a way to explicitly show that it depends on k.
- Fig 4: you cannot see some error bars, could you please improve this?

---

> ### Author Rebuttal · Authors · 2025-07-31
>
> We thank the reviewer for your valuable feedback and for carefully reading the contents of the appendix as well.
>
> > (Weakness 1-1) there are several methods for anomaly detection, ...
>
> We agree with the reviewer that deep learning-based anomaly detection is rapidly evolving with many promising methods. Our goal, however, is to establish a statistically principled framework for controlling false positives at a desired significance level. To clearly demonstrate the effectiveness of selective inference, we chose deep kNN—a simple, widely used, nonparametric baseline—as a proof of concept. In principle, our framework can extend to other methods, provided the selection event (i.e., \$\varepsilon\_{\boldsymbol{Y}} = \varepsilon\_{\boldsymbol{y}}\$ in Eq.(6)) is well characterized. We will add a discussion on this generalizability in the revised version.
>
> > (Weakness 1-2) Additionally, the evaluation is carried out in rather limited datasets. ...
>
> We agree that extending the evaluation to more diverse datasets would strengthen the empirical analysis. However, since our goal is not to develop SOTA anomaly detection methods, but rather to establish the theoretical foundation of selective inference for anomaly detection with guaranteed Type I error control (see Theorem 4.2), we chose the MVTec dataset, which is the most widely used benchmark in this field. We believe that experiments presented in the main paper and appendix --- including those on the MVTec dataset and ten tabular datasets shown in Figure 4 --- serve to empirically validate the correctness and robustness of our theoretical claims.
>
> > (Weakness 2) I think that while the authors clearly state the motivation, it would be great to discuss how this actually could help in anomaly detection.
>
> Our goal is not to develop a more accurate anomaly detection algorithm, but to offer a statistically principled framework for controlling false positives. Anomaly detection inherently involves a trade-off between false positives and false negatives, which is critical in high-stakes areas like medical diagnosis or industrial inspection. Our framework supports such controlled decision-making by computing p-values for individual test instances and making decisions based on a user-defined significance level (e.g., 0.05), thereby ensuring that the false positive rate stays below the target threshold (see Theorem 4.2). We will clarify this practical benefit in the revised version.
>
> > (Weakness 3) in the proof of Theorem 4.1, you have this: ...
>
> We apologize for the confusion caused by our typo. There is indeed the following typographical error in Appendix A.1, line 834:
>
> * Incorrect：$z = T(\boldsymbol{Y}) =||\boldsymbol{\eta}\_{\boldsymbol{Y}}^T \boldsymbol{Y}||\_2^2 = ||P \boldsymbol{Y}||\_2^2$
> * Correct：$z = \sqrt {T(\boldsymbol{Y})} = ||\boldsymbol{\eta}\_{\boldsymbol{Y}}^T \boldsymbol{Y}|| = ||P \boldsymbol{Y}||$
>
> With this correction, the equation pointed out by the reviewer becomes valid as follows:
>
> $$\mathbf{Y} = \mathcal{U}(\mathbf{y}) +  \mathcal{V}(\mathbf{Y}) \mathbf{z} =  \mathcal{U}(\mathbf{y}) + \lVert P\mathbf{Y}  \rVert \mathcal{V}(\mathbf{Y})$$
>
> We will correct this typo in the camera-ready version. We would like to emphasize that this correction does not affect the conclusions of the paper.
>
> > (Weakness 4) Additionally, I also think that the work could be explained better ...
>
> The statistic between lines 197 and 198,
> $$
> \mathcal{Q}\_{\boldsymbol{Y}} = \left( \frac{P\boldsymbol{Y}}{\lVert P\boldsymbol{Y} \rVert},\ (I\_{(n+1)d} - P)\boldsymbol{Y} \right),
> $$
> is derived from the sufficient statistic for the nuisance parameter. The concept of a nuisance parameter is fundamental in statistics; in this context, it refers to a parameter that is included in the null distribution but is not of direct inferential interest. To characterize the null distribution, it is necessary to eliminate the influence of the nuisance parameter. In our approach, this is achieved by conditioning on its sufficient statistic. This is a standard technique in the field of selective inference ; for more details, please refer to Section 3.1 of [1]. Although the formulation of $\mathcal{Q}$ may differ depending on the problem setting, the one we use is widely adopted in prior SI literature. For example, the quantities $\text{dir}\_{\mathcal{L}}(Y)$ and $\mathcal{P}\_{\mathcal{L}}^{\perp}Y$ in Section 2 of [2] correspond to our formulation of $\mathcal{Q}\_{\boldsymbol{Y}}$. In the camera-ready version, we plan to add further explanations about nuisance parameters and how they are treated in the selective inference literature.
>
> [1]https://arxiv.org/pdf/1410.2597
> [2]https://arxiv.org/pdf/1607.08211
>
> > (Weakness 5) finally, I think some qualitative examples in the main paper would be good.
>
> Regarding qualitative results, we currently present a variety of examples using the MVTec dataset in Appendix C.4. In the camera-ready version, we plan to consider moving a subset of these results into the main paper.
>
> > (minor weakness 1) L66: It's important to mention why quantifying statistical significance for such AD methods would be important
>
> Our proposed method enables rigorous control of the false positive rate in anomaly detection by quantifying statistical significance through hypothesis testing. This property is particularly important in high-stakes applications such as medical diagnostics and industrial quality control, where false positives may lead to substantial risks or costs. We intend to clarify this point more explicitly in the camera-ready version.
>
> > (minor weakness 2) L102: I think it's more self-supervised rather than semi-supervised, but I leave it to the author
>
> In the problem setting of anomaly detection, there are three possible scenarios: 1) when both normal and anomalous instances are available, 2) when only normal instances are available, and 3) when neither normal nor anomalous instances are available. The second scenario is sometimes referred to as semi-supervised anomaly detection in the literature, and we follow this convention in this paper.
>
> > (minor weakness 3) L123-124. Any ref about the heuristic method for choosing k?
>
> In this paper, we adopt a heuristic approach in which the value of $k$ is selected to maximize the anomaly score. This strategy has also been employed in previous studies on anomaly detection. For example, Eq (3) in reference [3] adopts the same heuristic.
>
> [3]https://www.sciencedirect.com/science/article/pii/S0375674221001540
>
> > (minor weakness 4) Fig 4: you cannot see some error bars, could you please improve this?
>
> As correctly noted, the error bars for the Bonferroni method in Fig 4 are barely visible. This is due to the highly conservative nature of the Bonferroni correction, which yields values so small that the corresponding error bars become negligible or vanish in the plot. We will address this issue in the camera-ready version by improving the visualization or providing clarification in the figure caption.
>
> > Other minor weaknesses
>
> Regarding the other minor weaknesses, we fully agree with the reviewers' comments and will properly address all of them in the revised version.
>
> >(Question 1)
>
> See our response to Weakness 3.
>
> >(Question 2)
>
> See our response to Weakness 2.
>
> >(Question 3)
>
> See our response to Weakness 1-2.
>
> >(Question 4)
>
> See our response to Weakness 1-1.
>
> >(Questions5). Isn't Eq (4) the same as Eq (5)? ...
>
> Eq (4):
> $$
> p = P\_{\rm H\_0}(T(\boldsymbol{X}^{\text{test}}, \{ \boldsymbol{X}\_i \}\_{i \in [n]} ) \geq T(\boldsymbol{x}^{\text{test}}, \{ \boldsymbol{x}\_i \}\_{i \in [n]})),
> $$
> can be rewritten using the same notation as in Eq (5) as follows:
> $$
> p = P\_{\rm H\_0} \left( \lVert \boldsymbol{\eta}\_{\boldsymbol{Y}} \boldsymbol{Y} \rVert\_2^2 \geq \lVert \boldsymbol{\eta}\_{\boldsymbol{Y}} \boldsymbol{y} \rVert\_2^2 \right).
> $$
> Here, $\boldsymbol{\eta}\_{\boldsymbol{Y}}$ is a random variable that depends on the deep $k$-NN inference, making the null distribution difficult to characterize analytically. In contrast, Eq (5) is given by:
> $$
> p = P\_{\rm H\_0} \left( \lVert \boldsymbol{\eta}\_{\boldsymbol{y}} \boldsymbol{Y} \rVert\_2^2 \geq \lVert \boldsymbol{\eta}\_{\boldsymbol{y}} \boldsymbol{y} \rVert\_2^2 \right),
> $$
> which corresponds to a incorrect but intuitive approach. This naive method neglects the randomness of $\boldsymbol{\eta}\_{\boldsymbol{Y}}$ by treating it as a fixed quantity $\boldsymbol{\eta}\_{\boldsymbol{y}}$. In this naive formulation, $\boldsymbol{\eta}\_{\boldsymbol{y}} \boldsymbol{Y}$ can be viewed as a linear transformation of the Gaussian variable $\boldsymbol{Y}$, and hence also follows a Gaussian distribution. This allows for the null distribution to be easily characterized.

---

> > ### Comment · Reviewer_ZerL · 2025-08-04
> >
> > Dear authors,
> >
> > I really appreciate your detailed response to my main criticisms and weaknesses. This time, I have to say I'm convinced with most of your answers, therefore I'll update my score from Borderline Reject to Accept. I'd really encourage the authors to incorporate all of these coments and update the sections they are mentioning in this comment, since I think the paper could be clearer by adding all of this (such as some qualitative results in the main paper, stronger SI motivation and typos). But to be fair, I agree with the answers. Well done! Good work and good rebuttal.

---

### Official Review · Reviewer_baZz · 2025-07-03

**Clarity:** 3
**Significance:** 3
**Originality:** 3
**Rating:** 4
**Confidence:** 3

**Summary:**

This paper studies semi-supervised anomaly detection, where the model is trained solely on normal data. It focuses on the existing "deep k-NN AD" method, which maps input data into a latent feature space using a pre-trained CNN, and then measures the
  distances between a test sample and the training points. While the paper does not propose a new detection algorithm, its main contribution lies in introducing a statistical framework to assess the significance of detections made by deep k-NN. Specifically, the authors formulate the AD process as a hypothesis testing problem and propose a selective p-value based on the selective inference framework. This approach aims to control the false positive rate and correct for selection bias in the AD setting.

**Questions:**

See Weaknesses

**Ethical Concerns:**

["NO or VERY MINOR ethics concerns only"]

**Limitations:**

See Weaknesses

**Paper Formatting Concerns:**

No formatting concerns.

**Quality:**

3

**Strengths And Weaknesses:**

Strengths:

This paper identifies a fundamental challenge in deep k-NN-based anomaly detection that computing the conventional p-value is intractable. To address this, the authors propose an alternative: a selective p-value derived from the selective inference framework, which is computationally feasible, and thus capable of effectively quantifying the false positive rate. This is a meaningful contribution, as most anomaly detection research (especially those on the theoretical side) tends to focus primarily on overall misclassification rate. However, in many real-world applications such as cybersecurity, controlling false positives is often more critical.  I believe this work makes a significant contribution to the anomaly detection literature.

Weaknesses:

1. I am not very familiar with the issue of "double-dipping", which the authors cite as a primary motivation for adopting selective inference to quantify statistical significance in deep k-NN AD. I understand that "double-dipping" refers to using the same dataset for both hypothesis selection and testing, why this issue cannot be addressed by simply splitting the dataset? Since this work specifically studies semi-supervised AD which trained only on normal data, the lack of labeled anomalies shouldn't prevent such a split. It would help the general audience if the authors could explain more precisely why double-dipping is unavoidable in this context, and whether the challenge is specific to the k-NN approach or applies more broadly.

2. Perhaps related to the above point, I understand that this work prioritizes minimizing the false positive rate, which is indeed critical in AD. However, the overall misclassification rate (including both false positives and false negatives) is also an important metric in AD research. I wonder if there is a trade-off between reducing the false positive rate and increasing the false negative rate in this setting? It would be helpful if the authors could comment on whether their method maintains a good balance, or if minimizing Type-I errors comes at the cost of missing true anomalies.

3. While reading the numerical experiment results, I was particularly struck by the consistently zero Type-I error achieved by the Bonferroni correction. It seems almost too good to be true. Can the authors provide a discussion comparing their proposed method to Bonferroni correction beyond what is shown in the experiments? Are there advantages that are not immediately visible from the reported metrics? Otherwise, based solely on the experimental results, one could argue that Bonferroni performs better than the proposed method, and that there may be little justification for adopting the new approach.

4. One thing of the synthetic data experiments left me confused. It appears that all the synthetic datasets were generated from Gaussian distributions. However, the authors mention that the Hotelling’s
$t^2$ test fails to control the Type-I error because it relies on the Gaussian assumption. If the data is Gaussian, shouldn't the test perform well under its assumed conditions?

Minor comment: There is a typo on line 18 on page 1. It should be "The" instead of "Tsehe".

---

> ### Author Rebuttal · Authors · 2025-07-31
>
> We thank the reviewer for your feedback.
>
> > (Weaknesses-1) I am not very familiar with the issue of "double-dipping", which the authors cite as a primary motivation for adopting selective inference to quantify statistical significance in deep k-NN AD. I understand that "double-dipping" refers to using the same dataset for both hypothesis selection and testing, why this issue cannot be addressed by simply splitting the dataset? Since this work specifically studies semi-supervised AD which trained only on normal data, the lack of labeled anomalies shouldn't prevent such a split. It would help the general audience if the authors could explain more precisely why double-dipping is unavoidable in this context, and whether the challenge is specific to the k-NN approach or applies more broadly.
>
> In our problem setting, anomaly detection is performed in the latent space of a pre-trained deep model. Our focus is on the *test phase*, where we input a test instance and evaluate its anomaly score based on its distance to nearest neighbors. Here, "double-dipping" arises because the same test instance is used twice: first to *select* whether it is anomalous (i.e., if its distance exceeds a threshold), and second to *test* whether that distance is statistically significant. This re-use violates the standard assumption of independence between selection and inference. Unlike other selective inference (such as feature selection), we cannot split the test instance (i.e., an image patch) for selection and inference. Our selective inference framework addresses this issue by conditioning on the selection of anomaly. This challenge is not specific to k-NN and applies broadly to anomaly detection relying on test-instance-derived statistics.
>
> > (Weaknesses-2) Perhaps related to the above point, I understand that this work prioritizes minimizing the false positive rate, which is indeed critical in AD. However, the overall misclassification rate (including both false positives and false negatives) is also an important metric in AD research. I wonder if there is a trade-off between reducing the false positive rate and increasing the false negative rate in this setting? It would be helpful if the authors could comment on whether their method maintains a good balance, or if minimizing Type-I errors comes at the cost of missing true anomalies.
>
> We fully agree that in anomaly detection, a trade-off between false positive and false negative rates is inevitable. In high-stakes applications such as medical diagnosis or industrial inspection, it is essential to explicitly control this trade-off to ensure reliable decision-making. Our contribution lies in developing a statistically principled framework that enables such control. Specifically, our method computes p-values for individual instances and makes decisions based on a user-specified significance level (e.g., 0.05). As theoretically guaranteed in Theorem 4.2, this approach allows for strict control of the false positive rate below the desired threshold. Once the false positive rate is controlled at the specified level, we assess the trade-off by analyzing the statistical power, i.e., 1 − false negative rate (see Figs. 3 and 4). In summary, our method is not designed merely to minimize false positives, but rather to support decision-making under explicitly guaranteed control of the false positive rate under the trade-off.
>
> > (Weaknesses-3) While reading the numerical experiment results, I was particularly struck by the consistently zero Type-I error achieved by the Bonferroni correction. It seems almost too good to be true. Can the authors provide a discussion comparing their proposed method to Bonferroni correction beyond what is shown in the experiments? Are there advantages that are not immediately visible from the reported metrics? Otherwise, based solely on the experimental results, one could argue that Bonferroni performs better than the proposed method, and that there may be little justification for adopting the new approach.
>
> The objective of statistical hypothesis testing is to control the Type I error rate (false positive rate) at a specified significance level, while maximizing statistical power as much as possible. Our experimental results indicate that the Bonferroni correction tends to be overly conservative, reducing the Type I error rate well below the specified significance level (0.05), which results in a substantial loss of statistical power (Fig. 4). This suggests that the Bonferroni correction may not be practical from the perspective of the aforementioned trade-off. In contrast, our proposed method offers the advantage of theoretically controlling the Type I error rate at the specified significance level while maintaining practically useful statistical power.
>
> > (Weaknesses-4) One thing of the synthetic data experiments left me confused. It appears that all the synthetic datasets were generated from Gaussian distributions. However, the authors mention that the Hotelling's $t^2$ test fails to control the Type-I error because it relies on the Gaussian assumption. If the data is Gaussian, shouldn't the test perform well under its assumed conditions?
>
> In our problem setup, we distinguish between the *signal component* $s$ and the *noise component* $\epsilon$ in the data, as described in Eq.(2). Hotelling's $T^2$ test assumes a fully parametric model in which the entire observation follows a multivariate Gaussian distribution. In contrast, our setting is *semi-parametric* in the sense that only the noise is assumed to follow a Gaussian distribution, while the signal is allowed to follow an arbitrary, potentially complex distribution. This semi-parametric setting is substantially less restrictive than that of the fully parametric setting. In the left panels of Figures 2 and 3, we simulate data under a parametric condition where the signal is Gaussian, and Hotelling's test performs as expected. In contrast, the right panels illustrate a semi-parametric condition where the signal is multimodal and non-Gaussian, leading to inflated Type-I error in Hotelling's method. We agree that this distinction may not have been sufficiently clear and will provide a more detailed explanation in the revised version.
>
> > Minor comment: There is a typo on line 18 on page 1. It should be "The" instead of "Tsehe".
>
> Thank you for pointing this out. We will make the necessary correction in the camera-ready version.

---

### Decision · Program_Chairs · 2025-09-17

**Decision:**

Accept (poster)

**Comment:**

The paper proposes a selective inference framework for quantifying statistical significance in deep k-NN anomaly detection. The contribution is well-motivated, addressing the lack of uncertainty quantification in distance-based anomaly detectors by providing p-values with finite-sample error guarantees. The work is technically sound, and the committee generally appreciated the articulation of the problem, the theoretical analysis, and the empirical demonstrations on synthetic, tabular, and image datasets.

At the same time, several concerns were raised, including the relatively narrow scope of the method (restricted to deep k-NN), reliance on Gaussian noise assumptions, limited theoretical novelty beyond adapting existing selective inference tools, and a somewhat modest empirical evaluation. The rebuttal was effective in clarifying some technical details in question (e.g., proof corrections, robustness under noise misspecification, etc.).

Overall, I share the view that this is a timely and valuable contribution to the anomaly detection literature. The authors are encouraged, when preparing the next revision, to better highlight the scope and unique contributions of this work in the broader landscape of selective inference for anomaly detection, and to further clarify limitations and generalizability. This will help position the paper more strongly and mitigate potential perceptions of overlap with related concurrent efforts.